# Environmental Impact on Harmful Species *Pseudo-nitzschia* spp. and *Phaeocystis globosa* Phenology and Niche

Stéphane Karasiewicz *[ID] and Alain Lefebvre *[ID]

IFREMER-Laboratoire Environnement Ressources de Boulogne-sur-Mer (PDG-ODE-LITTORAL-LERBL), Centre Manche Mer du Nord, 150 Quai Gambetta, 62200 Boulogne-sur-Mer, France
* Correspondence: stephane.karasiewicz@ifremer.fr (S.K.); alain.lefebvre@ifremer.fr (A.L.)

**Abstract:** Global environmental change modifies the phytoplankton community, which leads to variations in their phenology and potentially causes a temporal mismatch between primary producers and consumers. In parallel, phytoplankton community change can favor the appearance of harmful species, which makes the understanding of the mechanisms involved in structuring phytoplankton ecological niches paramount for preventing future risk. In this study, we aimed to assess for the first time the relationship between environmental conditions, phenology and niche ecology of harmful species *Phaeocystis globosa* and the complex *Pseudo-nitzschia* along the French coast of the eastern English Channel. A new method of bloom detection within a time-series was developed, which allowed the characterization of 363 blooms by 22 phenological variables over 11 stations from 1998 to 2019. The pairwise quantification of asymmetric dependencies between the phenological variables revealed the implication of different mechanisms, common and distinct between the taxa studied. A PERMANOVA helped to reveal the importance of seasonal change in the environmental and community variables. The Outlying Mean and the Within Outlying Mean indexes allowed us to position the harmful taxa niche among the rest of community and quantify how their respective phenology impacted the dynamic of their subniches. We also discussed the possible hypothesis involved and the perspective of predictive models.

**Keywords:** REPHY/SRN monitoring; spatial–temporal analysis; satellite data; biodiversity

## 1. Introduction

The phytoplankton community is an essential component of marine life as it supports higher trophic-level organisms, including those of economic importance, and it is directly impacted by climate and local hydrological conditions [1,2]. The apprehension of the impact of global environmental change on phytoplankton has led scientists to examine long-term or large-scale data series. They have discovered significant regime shifts caused by the modification of the phytoplankton community structure (i.e., the species number and their respective abundance) and by the alteration of bloom phenology in different ecosystems such as the North Pacific [3], North Atlantic [4], Baltic Sea [5] and North Sea [1,6]. The variation in phytoplankton phenology can potentially lead to temporal mismatch between primary producers and consumers, consequently affecting the populations at the highest trophic level as well as ecosystem functioning [6–8].

Changes to the phytoplankton community structure in itself can be alarming, with the spread and increasing impact of harmful algae blooms (HAB) [9–11]. Two groups of harmful algae exist [12]. The first group of algae produces toxins or harmful metabolites, such as toxins linked to wildlife mortality or human intoxication through seafood [12]. The second group is nontoxic but becomes harmful at high abundance [12]. The consequences of HAB are challenging for coastal resource management. The established strategies for preventing HAB negative impact vary considerably on location and on species involved [13]. The understanding of these phenomena has become an essential component for the development

of HAB prevention strategies. Although global climate change has led to increasing HAB events on a wide range of scales [10,14], it was thought best to address these issues using field observations that cover the same scales as the HAB impact [15,16].

In our study, the toxic group of harmful species will be represented by the genera *Pseudo-nitzschia* spp., which are globally distributed bacillariophytes in which some species, such as *Pseudo-nitzschia fraudulenta* and *P. australis*, are renowned for impacting ecosystems at different levels as well as creating economic losses [17,18]. The second group is represented by the genus *Phaeocystis*, which is one of the most globally distributed marine haptophytes [19]. Although nontoxic [20], it is classified as undesirable because three species (i.e., *P. globosa*, *P. pouchetii* and *P. antarctica*) can form large gelatinous colonies, creating impressive foam layers along beaches during bloom collapse [21]. In the eastern English Channel and southern bight of the North Sea, our study area, *P. globosa* blooms have destructive effects on benthic and pelagic ecosystems by causing deep ecosystem reorganization and further impacting fisheries and aquaculture, contributing to the negative perception of the environment by tourists [22–26]. Moreover, *P. globosa* can sometimes co-occur with *P. delicatissima* complex [27–29] which may be used as a solid substrate during its life stage transition [23,30]. In the studied area, three species of *Pseudo-nitzschia* were identified (*P. delicatissima*, *P. pungens*, *P. fraudulenta*) [31].

The processes involved in climate-induced change within the phytoplankton community, as well as its regime-shift, are misunderstood due to our restricted knowledge of phytoplankton species ecological niches [32]. A regime-shift (or sudden community shift) does not necessarily emerge from the transition between two stable states of the ecosystem [33]. Alternatively, it can be the outcome of interactions between climate-induced environmental change and the species' ecological niche [33]. In addition, it was also revealed that variation in the species ecological niche could cause differences in phytoplankton bloom intensity and successions [34,35]. It is, therefore, key to use an approach that would integrate the ecological niche concept within the study of the environmental conditions and biotic interaction effect on the phytoplankton community. However, the comprehension of the mechanisms involved in structuring the phytoplankton community ecological niches and their variations in seasonal community successions is an important but complex task to carry out [36,37].

Phenology is a valuable tool for detecting these changes. The recurrent cycles of the phytoplankton blooms that result from the interplay between physical and biological processes creates variations in annual timing and amplitude from one year to another [38]. Previous cases have already been shown with terrestrial and freshwater species [39,40], more recent studies in marine ecosystems have linked species phenological cardinal dates with the changing climate [41–43]. For HAB studies, Guallar et al. [44] used methods developed by Rolinski et al. [45] to characterize the blooms of *A. minutum*, adapted to the dataset and the species bloom features. Several phenological parameters have been determined and related to environmental variables [44]. In this study, a new methodology for characterizing bloom characteristics was developed to fit the time-series in which the taxa could have more than one bloom during the year (spring and autumn) and in which they are not necessarily bound to specific cardinal dates from a "man-made" calendar. As with Rolinski et al. [45], variables characterizing the blooms were determined after the detection of the blooms as well as additional parameters explained later in greater detail. The new bloom detection method along with extra variables describing each bloom can provide an insight into the mechanism driving harmful species phenology.

To obtain an insight into the mechanism influencing blooms, a wide temporal and spatial coverage of these events needs to be studied. Monitoring programs give the opportunity to conduct spatial and temporal studies, as they provide continuity through time and broad spatial coverage, as well as information on onset, termination, intensity and harmful outbreaks [11]. The French program for phytoplankton and phycotoxin monitoring (REPHY) and its local extended program the Nutrient Regional Survey (SRN), managed by IFREMER, have been recording occurrences of toxic and nontoxic species and envi-

ronmental variables since 1984 and 1992, respectively. Several studies have highlighted the valuable information gathered, including phytoplankton community shifts [28] or species niche characterization [46,47]. In addition to the in situ data collected, satellite imagery gives us the opportunity to collect further environmental information about the study area. The availability of satellite images offers great opportunity to fill the scientific gap concerning the extent and dynamics of HABs in the English Channel, by potentially offering full coverage of the area at high spatial and temporal resolutions [48]. Indeed, the availability of large-scale satellite images cannot replace in situ hydrological (e.g., nutrient concentration) and biological (e.g., concentration of phytoplankton species) analysis of water composition, as it only detects some of the water's physical parameters, such as its color. However, the combination of satellite observations with in situ data gives the possibility of establishing relationships between water color and total phytoplankton biomass in terms of chlorophyll.

The aim of our paper is to assess the triptych relationship between environmental condition, phenology and niche ecology of harmful species, *P. globosa* and the complex *Pseudo-nitzschia* along the French coast of the eastern English Channel. First, we detected and characterized the potential harmful blooms of the two taxa and assessed their potential mechanisms. Second, we identified the main spatial–temporal changes in environmental conditions and the phytoplankton community structure between 1998 and 2019. Finally, to understand the relationship between these changes, environmental conditions, phytoplankton community structure and phenology, the dynamics of the harmful taxa ecological niches were analyzed to quantify the potential effects of environmental variation and community changes on species phenology.

## 2. Materials and Methods

### 2.1. Dataset

IFREMER has been collecting information on the phytoplankton community and water chemical composition for the REPHY-SRN network, on a bimonthly basis, since 1984 and 1992, respectively [49,50]. The environmental variables measured and used in this study were water temperature (°C), salinity (measured using the Practical Salinity Scale), turbidity (NTU) and dissolved inorganic nutrient concentrations ($\mu$mol L$^{-1}$) which consist of ammonia ($NH_4^+$), nitrate plus nitrite ($NO_3^- + NO_2^-$ noted NO hereafter), silicate ($Si(OH)_4$, and phosphate ($PO_4^{3-}$).

In addition to REPHY hydrological parameters, daily photosynthetically active radiation (PAR, W m$^{-2}$), concentration in suspended particulate matter (SPM mg L$^{-1}$) and chlorophyll-*a* ($\mu$g L$^{-1}$) was derived from METEOSAT visible imagery [51,52] as they are not measured by REPHY. The stations located inside bays or estuaries made the direct assignation of satellite data difficult due to the unsuitability of the satellite imagery in such high turbidity areas. To overcome this limitation, a 6 km square zone was delimited around each sampling station as in [53]. The median of all available pixels within the zone was allocated to the station.

The phytoplankton abundance data were also extracted from the REPHY. Organisms were identified to the highest taxonomic level. Furthermore, phytoplankton experts identified and counted organisms whose size was <20 $\mu$m and smaller-size species if forming a chain structure or a colony. Smaller species were also counted if they were considered potentially toxic/noxious (e.g., *Chrysochromulina*, *Phaeocystis*). Further details about the phytoplankton physicochemical parameter sampling and processing are available in the literature [54–58]. Taxa with an occurrence of less than 5% relative to the total number of samples were excluded [59]. Taxa that were difficult to discriminate with optical microscopy were grouped into taxonomic units. This was the case for multiple species or even genera (e.g., *Pseudo-nitzschia seriata* complex, *Gymnodinium-Gyrodinium*); see Table 1. The method aims to limit the difficulties in discriminating between species or genera using optical microscopy. The taxonomic units guarantee homogeneity in a time-series between

different sampling sites [28]. Here, the study area covers the French coast of the eastern English Channel with 11 stations between 1998 and 2019 (Figure 1).

**Table 1.** List of the 47 phytoplanktonic taxonomic units used in the study. The code in bold are the harmful taxum studied.

| Phylum | Code | Species |
|---|---|---|
| Ochrophyta | Ast | *Asterionella* sp., *A. formosa, Asterionellopsis* sp., mainly *A. glacialis, Asteroplanus, A. karianus* |
| | Bac | *Bacillaria* sp., *Bacillaria paxillifer, Bacillaria paxillifera* |
| | Cer | *Cerataulina pelagica* |
| | Cha | *Chaetoceros* sp., *C. affinis, C. castracanei, C. curvisetus, C. danicus, C. debilis, C. decipiens, C. densus, C. didymus, C. fragilis, C. lorenzianus, C. peruvianus, C. protuberans, C. pseudocurvisetus, C. rostratus, C. socialis, C. socialis f. radians, C. subtilis, C. wighamii* |
| | Cos | *Coscinodiscus* sp., *Stellarima* sp. |
| | Cyl | *Cylindrotheca* sp., *Cylindrotheca closterium, Cylindrotheca gracilis, Nitzschia longissima, Hantzschia* sp. |
| | Dac | *Dactyliosolen* sp., mainly *D. fragilissimus* |
| | Dic | *Dictyocha* sp., mainly *Dictyocha fibula* |
| | Dip | *Diploneis* sp. |
| | Dit | *Ditylum* sp., mainly *D. brightwellii* |
| | Euc | *Eucampia* sp., mainly *E. zodiacus, Climacodium sp* |
| | Gui | *Guinardia* sp., *G. flaccida*, mainly *G. striata* and *G. delicatula* |
| | Lau | *Lauderia* sp., *L. annulata, Schroederella* sp. |
| | Lep | *Leptocylindrus* sp., *L. danicus, L. curvatus, L. minimus* |
| | Lic | *Licmophora* spp. |
| | Meu | *Meuniera* sp., *Meuniera membranacea* |
| | Nav | *Navicula* sp., *N. cryptocephala, N. gregaria, N. pelagica, Fallacia* sp., *Haslea* sp., *H. wawrikae, Lyrella* sp., *Petroneis* sp. |
| | Odo | *Odontella* sp., *O. aurita, O. granulata, O. mobiliensis, O. regia, O. Sinensis* |
| | Par | *Paralia sulcata* |
| | Pen | *Unidentified taxa of the Order Pennales* |
| | Pla | *Brockmanniella* sp., *Brockmanniella brockmannii, Plagiogramma* sp. |
| | Ple | *Pleurosigma* sp., *Gyrosigma* sp. |
| | **Pse** | *Pseudo-nitzschia spp* |
| | Rha | *Rhaphoneis* sp., *Delphineis* sp. |
| | Rhi | *Rhizosolenia* sp., *R. hebetata, R. imbricata, R. styliformis, R. setigera, R. setigera f. pungens, Neocalyptrella robusta* |
| | Ske | *Skeletonema* sp., mainly *Skeletonema costatum* |
| | Tha | *Thalassiosira* sp., *T. angulata, T. antarctica, T. gravida, T. levanderi, T. minima, T. nordenskioeldii, T. rotula, T. subtilis, Porosira* sp. |
| | Thao | *Thalassionema* sp., mainly *T. nitzschioides, Thalassiothrix* sp., *Lioloma* sp. |
| Myzozoa | Ale | mainly *Alexandrium minutum, A. margalefii, A. ostenfeldii, A. pseudogonyaulax* |
| | Amp | *Amphidinium* sp., *Amphidinium carterae, Amphidinium operculatum, Amphidinium crassum* |
| | DiP | *Diplopsalis* sp., *Diplopelta* sp., *Diplopsalopsis* sp., *Preperidinium* sp., *Oblea* sp. |
| | Gym | *Gymnodinium* sp., *G. catenatum, Gyrodinium* sp., *G. spirale* |
| | Het | *Heterocapsa* sp., *H. niei, H. triquetra* |
| | Kat | *Katodinium* sp. |
| | Pol | *Polykrikos* sp., *P. schwarzii* |
| | Pro | *Protoperidinium* sp., mainly *P. bipes, P. conicum, P. depressum, P. diabolum, P. longipes, P. steinii, P. pyriforme, Archaeperidinium minutum, Peridinium* sp., *P. quiquecorne* |
| | Proe | *Prorocentrum* sp., *P. arcuatum, P. balticum, P. cordatum, P. compressum, P. gibbosum, P. gracile, P. micans, P. triestinum* |
| | Pyr | *All Taxa of the Pyrocystaceae familly* |
| | Scr | mainly *Scrippsiella* sp., *Ensiculifera* sp., *Pentapharsodinium* sp., *Bysmatrum* sp. |
| | Tor | *Torodinium robustum* |
| Haptophyte | **Pha** | *Phaeocystis* sp. |
| Euglenophyte | Eug | *Euglena* sp., *Eutreptia* sp., *Eutreptiella* sp. |
| Cryptophyta | Cry | *All Taxa of the order Cryptomonadales* |
| Ciliophora | Mes | *Mesodinium* sp., mainly *Mesodinium rubrum* |
| | Cil | *Unidentified taxa of the Phylum Ciliophora* |
| Chlorophyta | Sce | *Scenedesmus* sp., mainly *Scenedesmus quadricauda, Desmodesmus communis* |
| | Chl | *Unidentified taxa of the Order Chlorophyceae* |

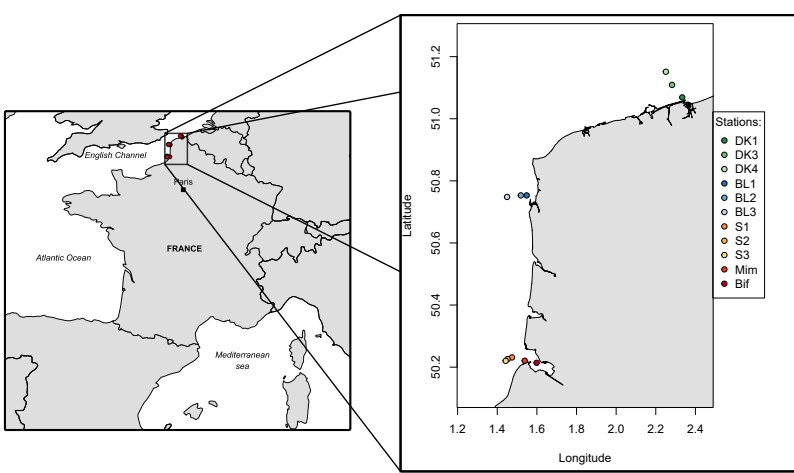

**Figure 1.** Map of the stations used in the study.

### 2.2. Community Diversity and Structure

Phytoplankton community diversity and equitability was estimated with the Shannon diversity index ($H'$) [60] and Pielou's evenness ($J'$) [61]. The indices were calculated with the diversity function available in the R vegan package [62]. The Shannon diversity index was calculated from the equation:

$$H' = -\sum p_i \ln(p_i) \tag{1}$$

where $p_i$ is the proportion of individuals for the $i$th taxonomic units. For an easier interpretation, the Shannon diversity index was converted into its numerical equivalent (also called the equivalent number of species) using the Hill (1973) formula:

$$H = exp(H) \tag{2}$$

Pielou's evenness was calculated as:

$$J' = H'/\ln(S) \tag{3}$$

where $S$ is the total number of species. $J'$ ranges from 0 to 1 with zero corresponding to no evenness and 1 meaning complete evenness. In addition to diversity and equitability, the community total number of taxonomic units $S$ was also used as the structure parameters of the community.

All statistical analysis was performed with the free statistical software R [63].

### 2.3. Spatial–Temporal Variation

Investigation into the temporal and/or spatial effect of the environmental conditions and community structure was done by performing a permutational multivariate analysis of variance (PERMANOVA [64]) on each dataset. PERMANOVAs are semi-parametric methods based on F-statistic estimation to support the partition of variance. The methods couple the robust statistical properties of rank-based nonparametric methods, with the possibility to test the effects of several factor interactions without the necessity for data normality [64]. PERMANOVAs performed in the study were based on Euclidean distance matrix and significantly tested with 9999 permutations. PERMANOVAs were done with the function `adonis2` available in the R vegan package [62]. The sampling stations were used as an explanatory factor for testing the spatial effect of datasets. For environmental and community variables, the temporal factors used were the months and years of the samples at which they were collected. Two temporal factors enabled the testing of monthly seasonal and inter-annual changes as a temporal effect. The month, the year and the stations factors were tested separately (3 factors), paired (3 different pairs of factors) and combined (all

together) for each variable of the environmental (10 variables) and community structure (3 variables) data given a total of $7 \times 13 = 91$ PERMANOVA analyses.

### 2.4. Niche Analysis: The Relationship between the Community and Environmental Condition

In this study, niche analysis was performed using Outlying Mean Index (OMI) analysis [65]. OMI analysis is a multivariate statistical analysis that estimates the species realized niche distribution of an entire community along environmental gradients [65]. For each taxonomic unit of the community, the analysis returns three niche parameters: OMI (basis of the analysis), tolerance (TOL) and residual tolerance (Rtol). The OMI parameter, or marginality, corresponds to the distance of the mean habitat used by a species to the mean habitat conditions of a hypothetical ubiquitous species [65]. A high marginality value for a realized niche implies that the taxonomic unit occurs in an atypical habitat compared to that with a low value, which occurs in a common habitat. The tolerance parameter (TOL), or niche breadth, is an estimation of the environmental range within which the species occurs. A taxonomic unit with a high tolerance value can be considered to be generalist, as it occurs in a broad range of environmental conditions. By contrast, a taxonomic unit with a low tolerance value would be considered to be specialist because it occurs in a limited range of environmental conditions [46]. The residual tolerance (Rtol) is a quantification of the variance lost after dimensional reduction. Rtol evaluates the suitability of the environmental variables used to define the species niche [65]. The niche parameters estimated with OMI analysis describes the response of harmful species to the environment. Their respective habitat was compared with the ones used by the rest of the phytoplankton community. OMI analysis provided an insight into the environmental variables defining the ecological niche of the harmful species.

### 2.5. Bloom-Detecting Algorithm

For phenology analysis, the time-series were split by station. They were then log-transformed $\log 10(x + 1)$ to compensate for the extreme values that the abundance can reach during HAB. For each time-series, the data were treated using a cubic smoothing spline algorithm [66] with the `smooth.spline` function from the package `stats` [63]. The smoothing parameter of the spline was selected with the Leave-One-Out Cross-Validation (LOOCV) method [67,68]. In addition to each fitted smooth spline, 95% confidence band intervals were calculated by bootstrapping [68]. After fitting a smooth spline to each time-series, an algorithm was developed to detect the species blooms.

The blooms were detected for each of the 11 time-series sampled at each station for the two harmful taxa, *Phaeocystis globosa* (Pha) and the complex *Pseudo-nitzschia* (Pse). The first step was to locate the high and low points of the curves. For each high point, the closest lowest points before and after were allocated to it. The combined low, high, and low points, which are characteristics of a hump, were considered to be a potential bloom. A detected hump was considered to be a bloom if:

1.  The high points were above the value of 4, which correspond to the log10 of 10,000 cells·L$^{-1}$. The threshold of 10,000 cells·L$^{-1}$ was used here in the study because, under this abundance threshold, the blooms of Pha and of Pse would not cause any HAB.
2.  The low points before and after the high points were inferior to 85% of the high point value. In this case, some humps can be merged as blooms can sometimes be bimodal.
3.  The merging of two humps would occur when the value of one of the lowest points do not fit the second condition. The merging of two humps cannot occur if the merging causes the increasing or decreasing phase of the bloom to be greater than 300 days.

These three conditions were necessary as they enable the extraction of the phenological bloom corresponding to HABs by defining the hump minimum abundance as well as the amplitude and shape. The number of blooms detected by stations and species is briefly described to obtain an overall picture of the differences between species and station.

*2.6. Phenological Variables*

For each bloom detected, 22 phenological variables were extracted (Figure 2). As with [44], the Maximum Abundance (MA) and the day of the Maximum Abundance (DMA) were directly obtained from the data by locating the DMA within the bloom period. To obtain the Day of the Bloom Start (DBS) and the Day of the Bloom End (DBE), we implemented the Extremum Surface Estimator (ESE) or Extremum Distance Estimator (EDE) when the ESE did not work, to find the inflection point of a convex/concave curve [69,70]. The ESE or EDE were performed using the `ese` and `ede` function from the `inflection` package. In the increasing phase, the inflection point detected is used as the DBS with its respective log10 abundance value at the start of the bloom (XO). Inversely, in the decrease phase, the inflection point detected corresponds to the DBE with its log10 abundance value (XE). As in [44], the Bloom Length (BL) was obtained by subtracting DBS from DBE; the Increasing Length (IL) by subtracting DBS from DMA, and the Decreasing Length (DL) by subtracting DMA from DBE. The Steepness Increase (SI) was estimated as follows:

$$SI = \log(MA/x0)/IL \tag{4}$$

and the following equation was used to calculate the Steepness Decrease (SD):

$$SD = \log(MA/xe)/DL \tag{5}$$

In addition to the nine phenological variables of [44], an extra 13 variables were estimated. [71] constructed the realized niche of phytoplankton using a fitness approach that consisted of extracting the point of Maximum Fitness (MF) from the smooth spline. The procedure consisted of calculating the net growth between two dates, therefore subtracting the abundance at time $t + 1$ with the values at time $t$. The MF would correspond to the highest difference in abundance between two dates. After locating the point of MF, the Date of Maximum Fitness (DMF) and the abundance at the DMF (XF) can be extracted. In the continuum of the Maximum Fitness idea, the same procedure can be repeated but on the decreasing phase of bloom. The point of Maximum Mortality (MM) would therefore correspond to the point at which the absolute difference between the abundance at time $t + 1$ and $t$ is the highest. As for MF, the Date of Maximum Mortality (DMM) and the abundance value at DMM (XM) can be extracted (Figure 2).

Similarly to the increasing and decreasing length (IL and DL), 5 additional variables were calculated: the onset (ONS), which is defined as the number of days between DBS and DMF; the climax (CLI) which represents the numbers of days between DMF and DMA (it is during the CLI period that the population has highest growth); the decline period (DEC), which is between the DMA and DMF, and corresponds to the time when the population starts to collapse; the end (END), which is between the DMM and DBE, and represents a period when the population stabilizes in abundance until the next bloom; and finally the harmful period (HAB), which is the period between the DMF and DMM, and considered to be when the taxa are at their most harmful due to population size coupled with potential capacity to produced toxins for *Pseudo-nitzchia*, and high biomass and consequently foam accumulation for *Phaeocystis*. The duration periods of the onset (ONS), the climax (CLI), the decline (DEC), the end (END) and of the harmful (HAB) period were used to define the bloom stages with which the subniches of the two harmful species were calculated (see the Subniche section).

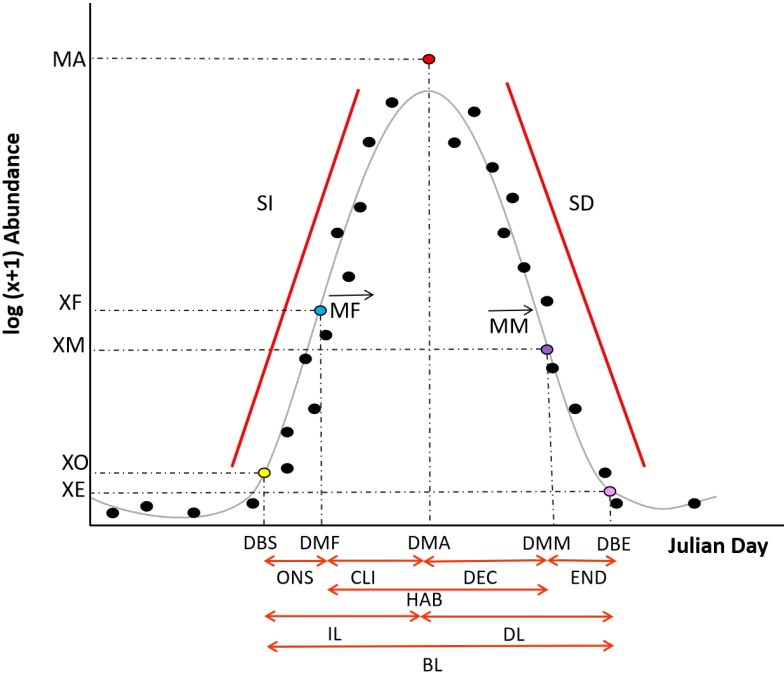

**Figure 2.** Summary of the 22 phenological variables extracted from each bloom detected in the time-series. DBS, Date of Bloom Start; XO, abundance at DBS; DMF, Date of Maximum Fitness; XF, Abundance at DMF; MF, Maximum Fitness, SI, Steepness Increase; MA, Maximum Abundance; DMA, Date of the Maximum Abundance; DMM, Date of Maximum Mortatlity; XM, Abundance at DMM; MM, Maximum Mortality; DBE, Date of the Bloom End; XE, Abundance at the DBE; SD, Steepness Decrease; ONS, ONSet phase duration; CLI, CLImax phase duration; DEC, DECline phase duration; END, ENDing phase duration; HAB, HArmful period; IL, Increasing Length; DL, Decreasing Length; BL, Bloom Length.

*2.7. Temporal Continuum*

Bloom detection does not separate the years, which means that a bloom can overlap between two years (e.g., autumnal bloom). These blooms can cause issues for the timing variables (e.g., DBS, DMF, DMA, MM and DBE) that were determined in Julian dates. For some blooms, the DBE was lower than their DBS, suggesting that the end of the bloom happened before it even started, which is impossible. In these cases, the phenological variables DBS, DMF, DMM and DBE would be transformed, if needed, to preserve a temporal continuum. To overcome the issue, each bloom would be allocated to the year of its DMA. The Julian day of the year would therefore start as 1 on 1 January and finish on 31 December of the same year as 365, or 366 for leap years. If the DBS and/or DMF happened before the 1 January of the year of the DMA, it would be a negative Julian day corresponding to the number of days before the 1 January DMA-IL-1. An extra day must be subtracted to account for the Julian date 0, which does not exist. Oppositely, if the DBE and/or DMM happened after the 31 December of the year of the DMA, it would be higher than 365 (or 366) as it would correspond to the number of days after the end of the year and can be calculated as DMA + DL.

*2.8. Phenological Analyses*

Comparison between Pha and Pse was done for each phenological variable with the Kruskal–Wallis Rank Sum Test [72]. The comparison between the species phenological variables revealed the differences or similarities in their respective bloom dynamics characteristics. The spatial–temporal variation in bloom phenology of the two species were considered in this section to study the potential shift in year-to-year phenology and/or spatial differentiation, but our exploratory analyses were inconclusive and therefore discarded.

The phenological variables extracted from the blooms of Pha and Pse were tested for dependence to understand the relationship between each pair of variables. The quantification of asymmetric dependence (qad) is a strongly consistent estimator of the dependence between a pair of variables $X$ and $Y$. The qad estimator is copula-based and scale-invariant, which quantifies the directed dependence measure $q(X, Y)$ by returning values between 0 and 1 [73]. Although the Pearson correlation coefficient assesses only linear and Spearman rank correlation monotonic relationships, qad can detect any kind of association. The qad estimator developed in [74] and performed with the R-package qad follows these six key properties:

1. $q(X, Y)$ can be estimated for all (continuous) random variables $X$ and $Y$, regardless of their parametric distribution.
2. $q(X, Y) \in [0, 1]$ (normalization).
3. $q(X, Y) = 0$ if and only if $X$ and $Y$ are independent (independence).
4. $q(X, Y) = 1$ if and only if $Y$ is a function of $X$, so we have $Y = f(X)$ and in this case there is a complete dependence or full predictability of $Y$ by $X$.
5. We do not necessarily have $q(X, Y) = q(Y, X)$ (asymmetry).
6. Scale changes do not affect $q(X, Y)$ (scale-invariance).

The temporal nature of the phenological variables makes, for some pairs of variables, two-way qad estimation ($q(X, Y)$ and $q(Y, X)$) impossible. Just as the DBE cannot happen before the DBS, the arrow of time prevents $Y$ having an effect on $X$ for some pairs of phenological variables. The one- or two-way qad estimation between the pairs of phenological variables are illustrated in Figure 3. To clarify the results and the different relationships, only the significant relationships were kept, but also those with a $q$ value > 0.5.

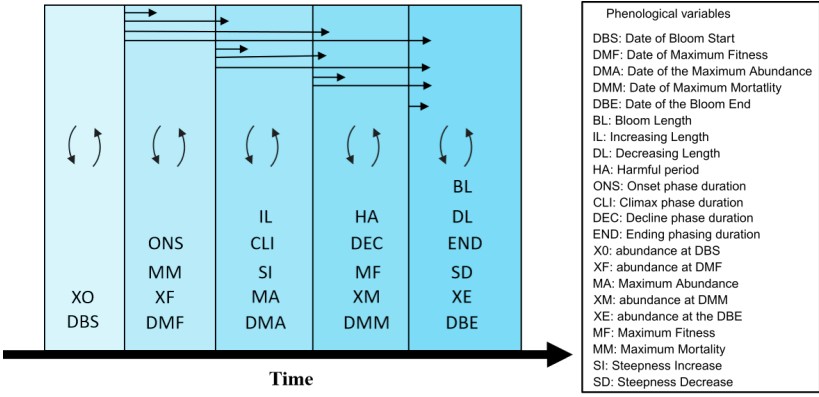

**Figure 3.** Summary of the relationship between phenological variables according to their respective timing of appearances. The blue boxes regroup the variables appearing at the same point in time. The straight black arrows represent the relationships between the variables of one box to another. The curves represent the relationships between the variables found within each box.

### 2.9. Subniche Calculation: Relating the Community and Environmental Condition to Phenology

To obtain a more detailed description of the harmful species phenology response to environmental and biotic variations, the bloom stages estimated previously—the onset (ONS), the climax (CLI), the decline (DEC), the end period (END), and the harmful (HA) period—were used to calculate the subniches. The Within Outlying Mean Index (WitOMI) was developed to decompose OMI analysis [75]. The index gives an estimation of the community niche shift and/or conservatism under different sub-environmental conditions (hereafter called a subset) that can be of temporal and/or spatial scale. Herein, the subsets were defined by the timing period of each bloom stage of the two harmful species *Phaeocystis globosa* and of the complex *Pseudo-nitzchia*. The WitOMI calculates the species niche occupation (from OMI analysis) within a subset of environmental conditions, which enable the study of the environmental conditions effect on the phenological subniche. Furthermore, the WitOMI decomposition of the niche into subniches contributes to the observation and

quantification of biological constraints B, and the unused part of the predicted subniche $S_P$ [75]. This property of the WitOMI enables the study of community variation regarding different phenological subniches. Overall, the WitOMI indexes give a quantification of the environmental and biotic pressure applied to the harmful species population at each stage of their bloom.

## 3. Results

### 3.1. Spatial–Temporal Variation

PERMANOVAs performed with the month as factor of variations were significant for all variables ($p < 0.01$) (Table 2). The variable where most of its variance can be explained by the monthly change was temperature ($R^2 = 0.897$). By contrast, the change in salinity ($R^2 = 0.022$) as well as SPM concentration ($R^2 = 0.045$) and turbidity ($R^2 = 0.080$) had a part of their respective variance explained by the changing month. H, the Shannon diversity index, and J, Pielou's evenness index, had similar proportions of their respective variance ($R^2 = 0.405$ and $0.401$ for H and J), which could be caused by monthly variation (Table 2).

**Table 2.** A summary of the PERMANOVAs performed on the environmental (SPM, suspended particulate matter; $NH_4^+$, ammonium concentration; NO, nitrate + nitrite concentration; PAR, photosynthetic active radiation; $PO_4^{3-}$, phosphate concentration; TEMP, temperature; SALI, salinity; $Si(OH)_4$, Silicate concentration; TURB, turbidity and CHLO, the concentration of chlorophyll-*a*)) and community diversity variables (H, Shannon diversity index; J, Pielou's evenness and S is the total number of species) with the month, year, and station as a single or combined (M: Months, Y: Years and S: Stations) explanatory factor. For each factor, the $R^2$ are given and the significance is represented by the * ($p$ value $< 0.05$).

| Variables | Months | Years | Stations | M:Y | M:S | Y:S | M:Y:S |
|---|---|---|---|---|---|---|---|
| **Environmental** | | | | | | | |
| CHLO | 0.238 * | 0.039 * | 0.039 * | 0.248 * | 0.028 | 0.03 | 0.239 |
| SPM | 0.045 * | 0.015 * | 0.06 * | 0.176 * | 0.07 * | 0.049 | 0.485 |
| $NH_4^+$ | 0.236 * | 0.057 * | 0.094 * | 0.152 * | 0.1 * | 0.049 * | 0.23 |
| NO | 0.456 * | 0.054 * | 0.122 * | 0.108 * | 0.045 * | 0.028 * | 0.126 |
| PAR | 0.638 * | 0.007 * | 0.002 | 0.095 * | 0.007 | 0.013 | 0.114 |
| $PO_4^{3-}$ | 0.251 * | 0.14 * | 0.022 * | 0.197 * | 0.016 | 0.067 * | 0.186 |
| SALI | 0.022 * | 0.164 * | 0.387 * | 0.108 * | 0.021 * | 0.05 * | 0.193 * |
| $Si(OH)_4$ | 0.41 * | 0.028 * | 0.122 * | 0.15 * | 0.073 * | 0.03 * | 0.153 * |
| TEMP | 0.897 * | 0.021 * | 0.002 * | 0.039 * | 0.011 * | 0.002 | 0.013 |
| TURB | 0.08 * | 0.011 * | 0.328 * | 0.129 * | 0.098 * | 0.045 * | 0.251 * |
| **Community** | | | | | | | |
| H | 0.405 * | 0.045 * | 0.004 | 0.155 * | 0.025 | 0.03 | 0.199 |
| J | 0.401 * | 0.039 * | 0.008 * | 0.16 * | 0.026 | 0.032 | 0.194 |
| S | 0.127 * | 0.233 * | 0.068 * | 0.149 * | 0.055 * | 0.052 * | 0.223 |

Annual variation was a significant explanatory factor for all variables (Table 2). The part of the variance explained by yearly variation was much lower for most variables. Only SALI and S, the total number of species, were better explained by annual variation than by monthly changes (Table 2). The variables with the highest $R^2$ was S with 0.233.

Spatial variations among the different stations were also a significant explanatory factor for most variables (Table 2). PAR and H were not statistically different between the stations ($p = 0.56$ and $0.103$ for PAR and H respectively). The variances of SALI ($R^2 = 0.387$) and TURB ($R^2 = 0.328$) were better explained by the station as a factor (Table 2).

A combination of the two temporal factors (M:Y in Table 2) was a significant combination of factors for all environmental and community variables. However, part of the variance explained by the combination of the temporal factors was lower than the monthly change alone (Table 2), as was the case for all the nutrient concentrations, TEMP, PAR, H and J. SALI was the only variable that has its variance better explained by annual changes than by combined temporal factors ($R^2$: $0.164 > 0.108$ for Years and M:Y respectively).

On the other hand, CHLO, SPM, TURB and S had a greater part of their respective variation explained when the two temporal factors were combined (Table 2).

The combined effect of the monthly change and stations was not significant for 5 variables (CHLO, PAR, $PO_4^{3-}$, H and J). For the significant variables, the part of their variance explained by M:S was relatively small, with 0.1 being the maximum for $NH_4^+$ (Table 2). For NO, SALI, $Si(OH)_4$ and S, the part of their variance explained by the combined effects of the changing months and stations was lower than if they were considered separately. The monthly changes better explained the variance of TEMP ($R^2$: 0.897 > 0.011) and $NH_4^+$ ($R^2$: 0.236 > 0.1) than M:S (Table 2). TURB was the only variable where its variation was better explained by spatial change than by M:S ($R^2$: 0.328 > 0.098). The combined effect of M and S was only beneficial to SPM ($R^2$ = 0.07), as its part variation explained was approximately the same as when only the spatial variation was considered ($R^2$ = 0.06) (Table 2).

Seven variables out of 13 were significant when the annual change was combined with spatial change (Table 2). For the significant variables, the part of their variations explained by Y:S was even smaller than for M:S, with the highest $R^2$ of 0.067 for $PO_4^{3-}$. $NH_4^+$, NO and SALI were better explained by the annual change and spatial variation individually than when combined (Table 2). The annual variation better explained $PO_4^{3-}$ ($R^2$: 0.14 > 0.067) and S ($R^2$: 0.233 > 0.052) variation than Y:S. On the other hand, the spatial change was a better explanatory factor for $Si(OH)_4$ ($R^2$: 0.122 > 0.03) and TURB ($R^2$: 0.328 > 0.045) than Y:S (Table 2).

The results from the combined explanatory factor of monthly change, annual variation and spatial differences were significant for only three environmental variables (Table 2). TURB had the highest $R^2$ (0.251) but the variance was better explained by the stations ($R^2$ = 0.328). A similar trend was observed for SALI ($R^2$: 0.193 < 0.387) (Table 2). Finally, $Si(OH)_4$ was better by the monthly change than by the combination of M:Y:S ($R^2$: 0.153 < 0.41).

SALI (0.387 > 0.164 and 0.022) and TURB (0.328 > 0.011 and 0.08) were the two variables that expressed a greater spatial variation than temporal variation (Table 2). The best explanatory temporal factor was the changing month, as it explained the greatest part of variance for CHLO, $NH_4$, NO, PAR, $PO_4^{3-}$, $Si(OH)_4$, TEMP, H and J. S was the only variable that was better explained by the annual change. The variations in SPM concentration were low, regardless of the explanatory factor ($R^2$ = 0.045 with months, $R^2$ = 0.015 with years and $R^2$ = 0.060 with stations) (Table 2). For most of the variables, the combined pairs of explanatory factors (M:Y, M:S, Y:S) were not better at describing the variations of the different variables (Table 2). Only the variations of CHLO ($R^2$ = 2.48) and SPM ($R^2$ = 0.176) were best described by the combined temporal factors than by any other combination or single factors. The results suggest a higher temporal than spatial impact on the data, as 9 variables over 13 had their maximum $R^2$ explained by M and Y than by S. Furthermore, the monthly change had a much higher impact than annual variations (8 out of 9 had their maximum $R^2$ explained by M). The spatial difference between stations only affected SALI and TURB, which implies potential salinity and turbidity gradients despite a rather homogeneous environment and community structure.

### 3.2. Niche

The Outlying Mean Index (OMI) analysis was significant ($p < 0.001$) and revealed that the 47 taxa had a significant realized niche (see Table 3). The first two principal components comprise 86.8% of the total projected inertia (70.48% for PC1 and 16.32% for PC2 labeled OMI1 and OMI2 hereafter) (Figure 4A).

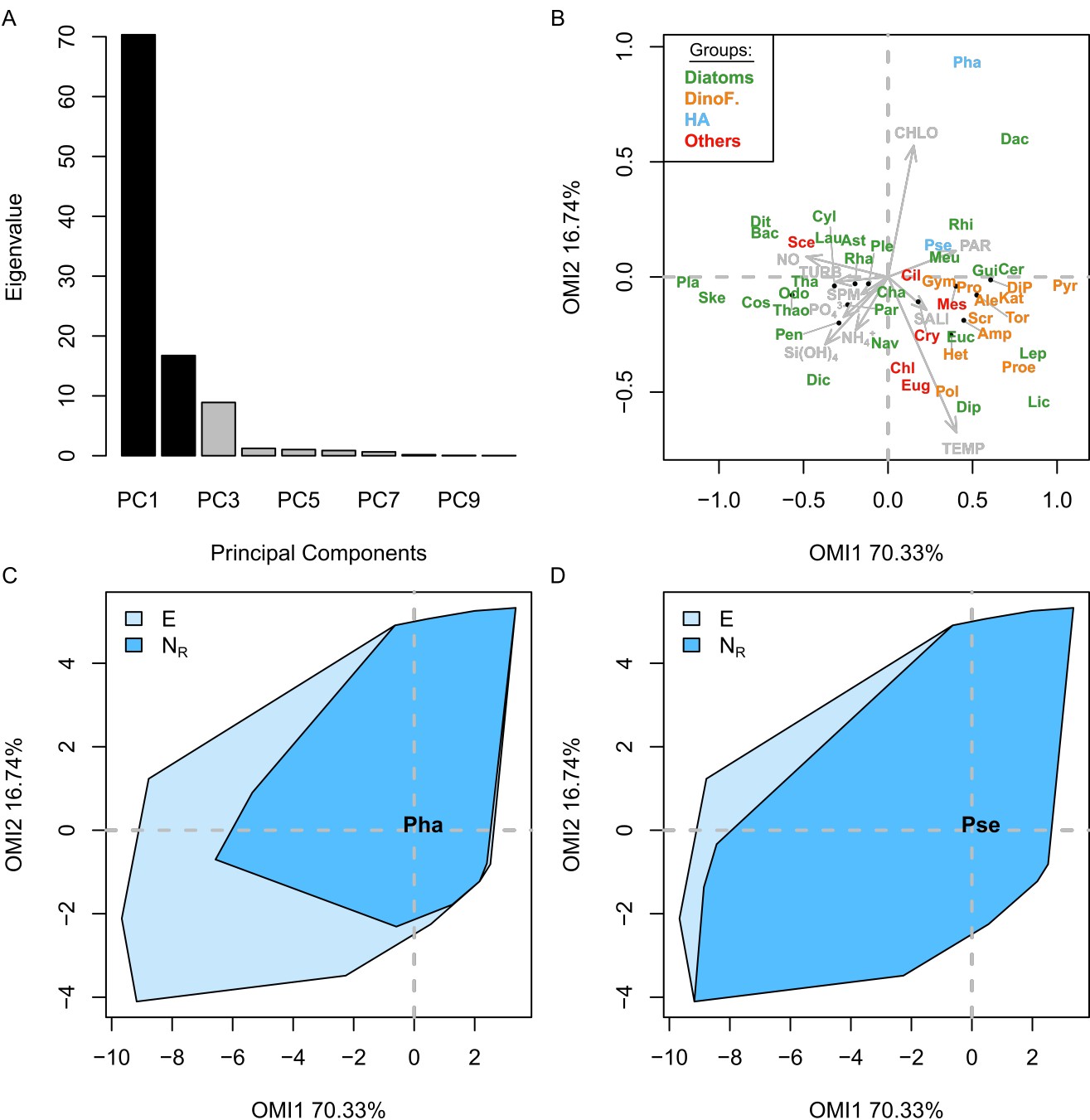

**Figure 4.** Results of the OMI analysis. (**A**) Bar plot representing the eigenvalue in percentages of the total sum. The black bars are the chosen factorial axis PC1 and PC2, which are named OMI1 and OMI2 for the other graph. (**B**) Representation of the significant species' realized niche positions on the first two factorial axes with the canonical weights of environmental variables (SPM, suspended particulate matter; $NH_4^+$, ammonium concentration; NO, nitrate + nitrite concentration; PAR, photosynthetic active radiation; $PO_4^{3-}$, phosphate concentration; TEMP, temperature; SALI, salinity; $Si(OH)_4$, silicate concentration; TURB, turbidity and CHLO, the concentration of chlorophyll-*a*). The two niche positions of the two taxa of interest, the complex *Pseudo-nitzchia* (Pse) and *Phaeocystis globosa* (Pha), are represented by the blue labels, respectively. The main groups of phytoplankton are represented by the different colors (Diatoms in green, Dinoflagellate in orange, and other groups in red). (**C**,**D**) are the representations of the realized niche, $N_R$, position (black label) and breadth (blue polygon) found within the environmental space E (light blue polygon) for *Phaeocystis globosa* (Pha) and for the complex *Pseudo-nitzchia* (Pse) respectively.

The OMI plan was divided into two sections by the 10 environmental variables (Figure 4B): on the left side, the estuarine and winter-like conditions, and on the right side the more open-water and summer-like conditions. On the positive side of the first OMI axis, the OMI plan was characterized by the negative correlation between two groups of variables made of the positive correlation between the concentrations in nutrient ($NH_4^+$, $PO_4^{3-}$, and $Si(OH)_4$) and suspended particulate matter (SPM) for one and by the positive interaction between water turbidity (TURB) and nitrate + nitrite concentration (NO) (Figure 4B). The positive side of OMI1 seemed split into two regions. The lower part was explained by a positive correlation between temperature (TEMP) and salinity (SALI), whereas the top part of the figure appears to be defined by the positive correlation between irradiance (PAR) and the concentration of chlorophyll-*a* (CHLO) (Figure 4B). TEMP and SALI were negatively correlated with CHLO and to the group made of TURB and NO. On the other hand, the TEMP-SALI duo were not or poorly related to PAR on the negative side, and to the group made of SPM, $NH_4^+$, $PO_4^{3-}$, and $Si(OH)_4$ concentrations on the positive side (Figure 4B). Furthermore, CHLO and the NO-TURB duo had a weak correlation.

The community appeared to be well scattered across the bottom part of the OMI plan. The left side of the OMI plan was dominated by diatoms, with the presence of Sce. The environment dominated by the realized niche position of the diatoms was characterized by high amounts of nutrients and SPM concentration and with high water turbidity, low irradiance and chlorophyll-*a* concentration (Figure 4B). On the left side of the OMI plan, there was no dominance of any group but all the dinoflagellate taxa, the two harmful taxa, and most of the other groups were found there. The dinoflagellates appeared to have a greater affinity for higher light intensity with lower turbidity and reduced nutrient and SPM concentrations (Figure 4B). Among the dinoflagellates, their realized niche positions appeared to be distributed diagonally, along the temperature and salinity gradients downwards. Similarly, the other groups of taxa were distributed vertically along a gradient defined by temperature–salinity and chlorophyll-*a* concentration (Figure 4B). The two taxa, Pha (harmful) and Dac (diatom), appear to have respective marginal realized niche position (OMI of 1.00 and 1.07 for Dac and Pha respectively) (Table 3).

Dac and Pha had a preference for a habitat made of high solar irradiance, with lower temperature, salinity, water turbidity, nutrient and suspended particulate matter concentrations (Figure 4B). Both Dac and Pha were correlated with a high concentration of chlorophyll-*a* because they were the taxa responsible for it. Pse appears to have preferred a more average kind of habitat (OMI:0.111) characterized by higher concentration of nutrients in warmer waters and with a community producing a lower concentration of chlorophyll-*a* and (Figure 4B).

The realized niche of Pse was larger than the one of Pha (Figure 4C,D). The larger realized niche was confirmed by the higher tolerance of Pse (Tol:2.8) over Pha (Tol:1.7) (Table 3). Pse appears to have preferred a more average kind of habitat (OMI:0.111) characterized by a higher concentration of nutrients in warmer waters and low concentration of chlorophyll-*a* in the community (Figure 4B). Pha appears to be more specialized for habitat conditions with high production of chlorophyll-*a* (Figure 4C), whereas Pse was more suited to environments with higher nutrient concentration (Figure 4D). In comparison to the rest of the community, Pha (OMI:1.07) had a high marginality whereas Pse had a small (OMI:0.111) (Table 3). The linear regression performed on the relationship between the taxa marginality and tolerance revealed some contrasting trends within the community (Figure 5). In general, for all taxa found within the community, the more uncommon its preferential habitat, the greater its tolerance (the black dashed line Figure 5). A similar and more compelling trend was also found for the diatoms (the green dashed line Figure 5). Two diatoms had lowest and highest values for marginality and tolerance. Mel had the highest OMI (2.326) and TOL (7.439) parameter in opposition of Cha had the lowest (OMI:0.01 and TOL: 0.709) for the same parameters (Table 3). In opposition to the common trend, the dinoflagellates seemed to have a higher tolerance when their respective preferential habitat was more common (the orange dashed lines Figure 5). For the group dinoflagellates,

Gym has the lowest marginality (OMI: 0.102) value with the highest tolerance (TOL: 2.836). On the other hand, Proe has the highest OMI value (0.796) and Pol the lowest TOL value (1.114) (Table 3).

**Table 3.** Niche parameters calculated with the OMI analysis for 47 taxa including *Phaeocystis globosa* (Pha) and the complex *Pseudo-nitzschia* spp (Pse) found in bold. The parameters are Inertia, marginality (OMI), the tolerance (Tol) and residual tolerance (Rtol). The *p* values were calculated with 1000 permutations; see methods for further details.

| Phylum | Code | Inertia | OMI | Tol | Rtol | *p* Value |
|---|---|---|---|---|---|---|
| Ochrophyta | Ast | 10.606 | 0.166 | 2.144 | 8.297 | 0.001 |
| | Bac | 12.439 | 0.558 | 3.812 | 8.069 | 0.001 |
| | Cer | 9.719 | 0.586 | 1.653 | 7.48 | 0.001 |
| | Cha | 8.764 | 0.011 | 0.74 | 8.014 | 0.001 |
| | Cos | 13.068 | 0.728 | 4.205 | 8.135 | 0.001 |
| | Cyl | 11.327 | 0.111 | 3.978 | 7.239 | 0.001 |
| | Dac | 8.983 | 1.007 | 1.412 | 6.565 | 0.001 |
| | Dic | 10.308 | 0.458 | 1.8 | 8.049 | 0.001 |
| | Dip | 8.176 | 0.688 | 0.829 | 6.659 | 0.001 |
| | Dit | 11.105 | 0.653 | 2.654 | 7.798 | 0.001 |
| | Euc | 7.967 | 0.349 | 0.921 | 6.696 | 0.001 |
| | Gui | 8.138 | 0.337 | 2.072 | 5.729 | 0.001 |
| | Lau | 9.58 | 0.229 | 2.157 | 7.194 | 0.001 |
| | Lep | 7.963 | 0.859 | 1.411 | 5.693 | 0.001 |
| | Lic | 6.473 | 1.164 | 1.415 | 3.894 | 0.001 |
| | Meu | 7.702 | 0.174 | 1.861 | 5.667 | 0.001 |
| | Nav | 10.727 | 0.105 | 1.095 | 9.527 | 0.001 |
| | Odo | 12.803 | 0.465 | 3.911 | 8.426 | 0.001 |
| | Par | 10.781 | 0.07 | 3.673 | 7.039 | 0.001 |
| | Pen | 11.198 | 0.142 | 3.02 | 8.036 | 0.001 |
| | Pla | 15.986 | 1.516 | 4.866 | 9.604 | 0.001 |
| | Ple | 10.957 | 0.032 | 2.808 | 8.117 | 0.001 |
| | **Pse** | 8.871 | 0.112 | 2.797 | 5.962 | 0.001 |
| | Rha | 9.898 | 0.063 | 2.261 | 7.574 | 0.001 |
| | Rhi | 8.738 | 0.253 | 2.268 | 6.217 | 0.001 |
| | Ske | 12.15 | 1.156 | 3.77 | 7.225 | 0.001 |
| | Tha | 11.09 | 0.303 | 3.572 | 7.215 | 0.001 |
| | Thao | 11.728 | 0.333 | 4.076 | 7.319 | 0.001 |
| Myzozoa | Ale | 6.473 | 0.369 | 1.751 | 4.353 | 0.001 |
| | Amp | 7.378 | 0.341 | 1.378 | 5.659 | 0.001 |
| | DiP | 8.121 | 0.398 | 1.86 | 5.863 | 0.001 |
| | Gym | 8.745 | 0.096 | 2.828 | 5.821 | 0.001 |
| | Het | 8.951 | 0.313 | 1.686 | 6.951 | 0.001 |
| | Kat | 7.62 | 0.548 | 1.795 | 5.277 | 0.001 |
| | Pol | 8.877 | 0.466 | 1.072 | 7.339 | 0.001 |
| | Pro | 7.671 | 0.198 | 2.414 | 5.059 | 0.001 |
| | Proe | 7.261 | 0.78 | 1.662 | 4.819 | 0.001 |
| | Pyr | 5.945 | 1.159 | 1.058 | 3.727 | 0.001 |
| | Scr | 8.705 | 0.376 | 2.147 | 6.183 | 0.001 |
| | Tor | 7.429 | 0.476 | 1.782 | 5.171 | 0.001 |
| Haptophyta | **Pha** | 8.485 | 1.102 | 1.722 | 5.661 | 0.001 |
| Euglenozoa | Eug | 10.791 | 0.304 | 1.042 | 9.445 | 0.001 |
| Cryptophyta | Cry | 9.207 | 0.049 | 2.112 | 7.045 | 0.001 |
| Ciliophora | Mes | 6.876 | 0.178 | 1.681 | 5.017 | 0.001 |
| | Cil | 9.567 | 0.046 | 2.459 | 7.062 | 0.001 |
| Chlorophyta | Sce | 19.656 | 1.527 | 3.548 | 14.582 | 0.001 |
| | Chl | 11.366 | 0.233 | 1.297 | 9.836 | 0.001 |

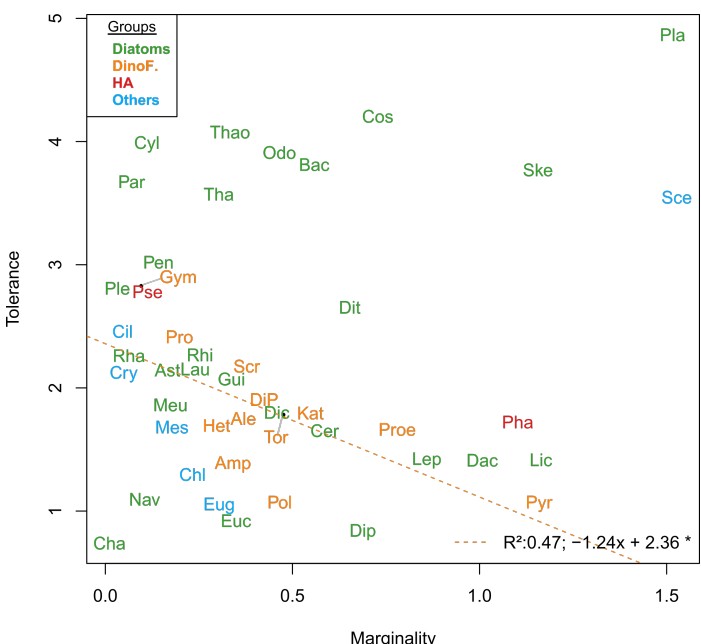

**Figure 5.** Biplot summarizing the relationship between the marginality (OMI parameter) and the tolerance (TOL) of the community's realized niches. The dashed lines represent the linear regression performed over OMI and TOL values over all taxa (black), the diatoms (green), and the dinoflagellates (orange). The * in the legend shows the significance of the linear regression ($p < 0.05$).

### 3.3. Phenological Data

The algorithm detected a total number of 363 blooms—159 for the complex *Pseudonitzchia* (Pse) and 204 for *Phaeocystis globosa* (Pha)—between 1998 and 2019 (Table 4). All the time-series with the fitted smooth splines can be found in the Supplementary Materials Figures S1 and S2 for Pha and Pse, respectively. A higher number of blooms was detected for Pha than for Pse with the exception of DK3 (see Table 4). The lower number of blooms detected for Pse was 12 at three distinct stations, DK4, BL3 and S2. By contrast, the higher number of blooms detected was 17 for Pse for the time-series of Mim, Bif and DK3. Concerning Pha, the higher number of blooms detected was 22 for station S3 and the lowest was 16 at station DK3 and DK4. The Bif station was the station with the highest number of blooms detected—38 in total (21 and 17 for Pha and Pse, respectively) (Table 4). On the other hand, the lowest number of blooms detected was of 28 at the DK4 station (16 and 12 for Pha and Pse, respectively) (Table 4). From these blooms, 22 phenological variables were extracted with their respective values summarized in Figure 6.

The detailed results of the Kruskal–Wallis Rank Sum Test performed on each phenological variable for the two species can be found in Supplementary Materials (Table S1). The median Date of the Bloom Start (DBS) and Date of Maximum Fitness (DMF) were not significantly different for the two species (DBS: 15 for Pse and 18 for Pha; DMF: 70 for Pse and Pha) (Figure 6A). The median Date of Maximum Bloom (DMA), Maximum Mortality (DMM) and Bloom End (DBE) were significantly earlier in the year for Pha than for Pse (DMA: 143 for Pse and 112 for Pha; DMM: 199 for Pse and 155 for Pha; DBE: 264 for Pse and 198 for Pha) and with a narrower timing window (Figure 6A). The lengths of the different bloom stages were significantly longer for Pse, with a higher fluctuation than for Pha (BL: 246 and 186 days; IL: 128 and 94 days; DL: 120 and 86 days; HA: 128 and 90 days; ONS: 51 and 43 days; CLI: 68 and 46 days; DEC: 128 and 90 days; END: 49 and 43 days for Pse and Pha respectively) (Figure 6B). The abundance value at the start of the bloom (X0) was significantly lower for Pha (100 and 1000 cells·L$^{-1}$ for Pha and Pse, respectively). Then, an equal value in abundance at maximum fitness (XF: 10,000 cells·L$^{-1}$) was attained between Pse and Pha, and therefore not significantly different (Figure 6C). The Maximum Abundance (MA) during the blooms were significantly higher for Pha ($10^7$ cells·L$^{-1}$) than

for Pse ($10^6$ cells·$L^{-1}$) and both dropped to approximately 10,000 cells·$L^{-1}$ abundance at the point of maximum mortality (XM) but were significantly different from one another. Finally, the abundance value at the end of the bloom (XE) returned to the lower values at which they started (100 and 1000 cells·$L^{-1}$ for Pha and Pse, respectively) (Figure 6C). The maximum fitness and mortality were significantly higher and with greater variation for Pha than for Pse ($10^{4.4}$ and $10^{3.2}$ cells·$L^{-1}$ for Pha and Pse, respectively) (Figure 6D). The slopes of Pha's blooms, during the increasing and decreasing phase of the bloom, were significantly steeper than the ones of Pse (SI: 0.11 and 0.05 days$^{-1}$; SD: −0.13 and −0.05 days$^{-1}$ for Pha and Pse, respectively), despite wider variations in steepness (Figure 6E).

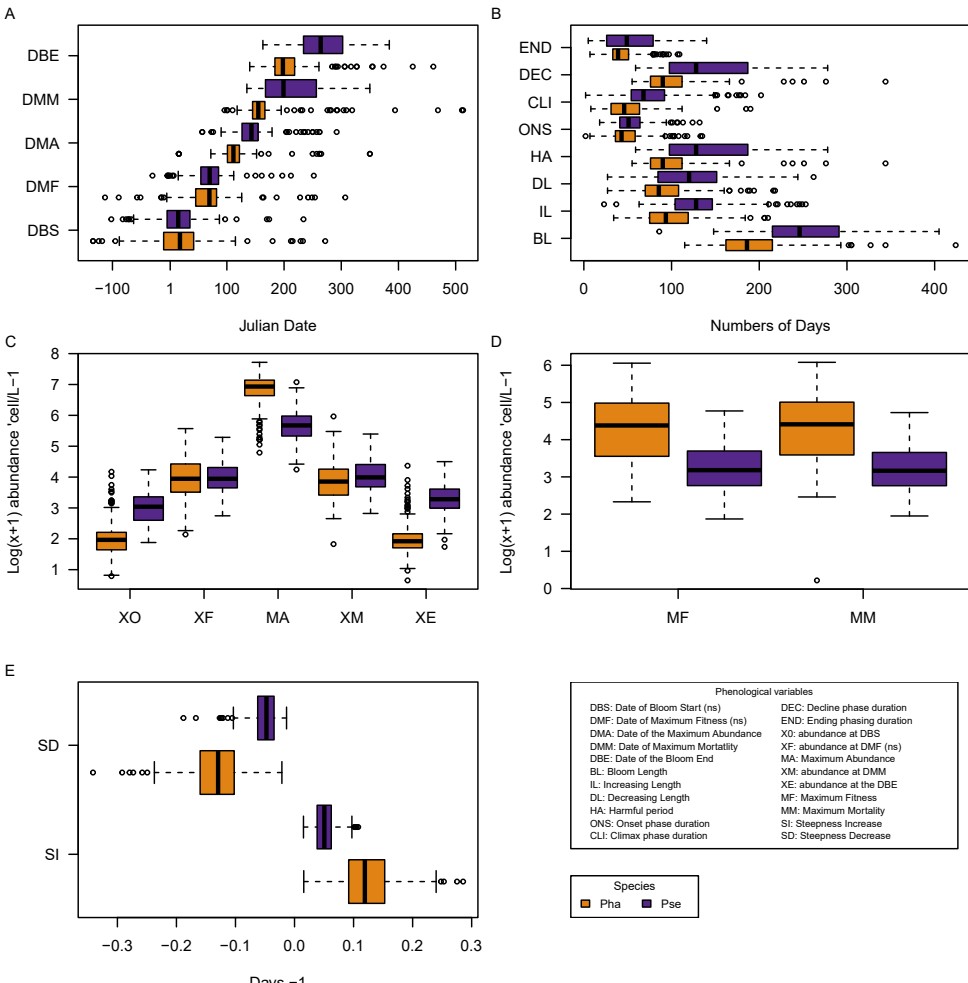

**Figure 6.** Boxplot summarizing the variations of the different phenological variables for the complex *Pseudo-nitzschia* (Pse) in purple and for *Phaeocystis globosa* (Pha) in orange. The middle bars in the boxes represent the median (second quartile); the end boxes are the first and third quartile limiting the middle half of the distribution of the variable; and the dotted line is the data statistical dispersion, calculated as 1.5 times the interquartile distance. Points beyond the dotted lines are possible outliers. (**A**) groups the summary of the dates for the different bloom stages according to the temporal continuum. (**B**) illustrates the variations in day length of the 5 different bloom stages, as well as the bloom increasing and decreasing phase with the total length. (**C**) shows the differences in abundance for each date of the bloom stages. (**D**) confronts the variations of Maximum Fitness and Mortality. (**E**) shows the variations in the bloom steepness. The phenological variables with (ns) in the legend are the ones which were not significant during the species comparison with the Kruskal–Wallis Rank Sum Test.

**Table 4.** Summary of the number of blooms detected by stations for the two species, *Phaeocystis globosa* (Pha) and the complex *Pseudo-nitzchia* (Pse). For station locations, see Figure 1.

| Station | Pse | Pha | Sum |
|---|---|---|---|
| DK1 | 14 | 19 | 33 |
| DK3 | 17 | 16 | 33 |
| DK4 | 12 | 16 | 28 |
| BL1 | 16 | 20 | 36 |
| BL2 | 14 | 19 | 33 |
| BL3 | 12 | 17 | 29 |
| S1 | 13 | 19 | 32 |
| S2 | 12 | 17 | 29 |
| S3 | 15 | 22 | 37 |
| Mim | 17 | 18 | 35 |
| Bif | 17 | 21 | 38 |
| All | 159 | 204 | 363 |

*3.4. Relationships between Phenological Variables*

The results of pairwise quantification of asymmetric dependencies between the phenological variables for the two taxa are reviewed in Figure 7. Figure 7A,B characterizes the blooms of Pha and Pse, respectively. The coefficients and significance can be found in Figure S3. From the 22 variables, 16 and 18 variables had correlations with others for Pha and Pse, respectively. For the two taxa, the variables DMA, ONS, END and XE were not related to any other variables. In addition for Pha, XO and MA were also isolated (Figure 7). The two taxa had 22 similar correlations between variables (black arrows Figure 7), with an additional 6 and 12 unique correlations for Pha and Pse represented by the orange and purple arrows, respectively (Figure 7). In total, Pha has 28 links between variables (Figure 7A) and Pse has 34 links between its variables (Figure 7B).

The common features between the two taxa were the effect of the DBS on DMF (0.64 and 0.63 for Pha and Pse; Figure S3) and the influence of DMM on DBE (0.57 and 0.61 for Pha and Pse; Figure S3). Interestingly, there was no continuity in the dependence between the Dates variables. Furthermore, it is worth noting that there is no direct correlation between the date variables (DBS, DMF, DMA, DMM and DBE) and their respective abundance variables (XO, XF, MA, XM and XE). Additionally, the two taxa have in common the symmetric dependencies between DBE and DL (0.5 and 0.6 for Pha and Pse, respectively) and between SD and DL (0.7 and 0.6 for Pha and Pse, respectively; Figure S3) (Figure 7). Another common pattern between the two taxa was the strong symmetric dependencies between HA and DEC bloom phases (0.93 and 0.94 for Pha and Pse, respectively; Figure S3) and separately affecting the BL (0.584 and 0.56 for Pha and Pse, respectively; Figure S3) (Figure 7). The interdependence between IL and SI appeared stronger in the case of Pha (0.77) than Pse (0.53) (Figure S3). The correlation between the CLI and IL (0.6 and 0.65 for Pha and Pse respectively) suggested that IL was more reliant on the length of CLI and vice versa. In addition, SI had a direct dependence on CLI, adding to the importance of the climax (CLI) phase during the increasing phase of the bloom (Figure 7). Finally, the interdependence between MF and XF (0.66 and 0.6 for Pha and Pse, respectively) appeared to have a link to the interdependence between MM and XM (0.6 and 0.5 for Pha and Pse, respectively) (Figure 7). The MF strongly influenced MM (0.92 and 0.86 for Pha and Pse, respectively) and to a lesser extent XM (0.66 and 0.52 for Pha and Pse, respectively). Additionally, XF was related MM (0.69 and 0.60 for Pha and Pse, respectively) but does not appear to affect XM (Figure 7).

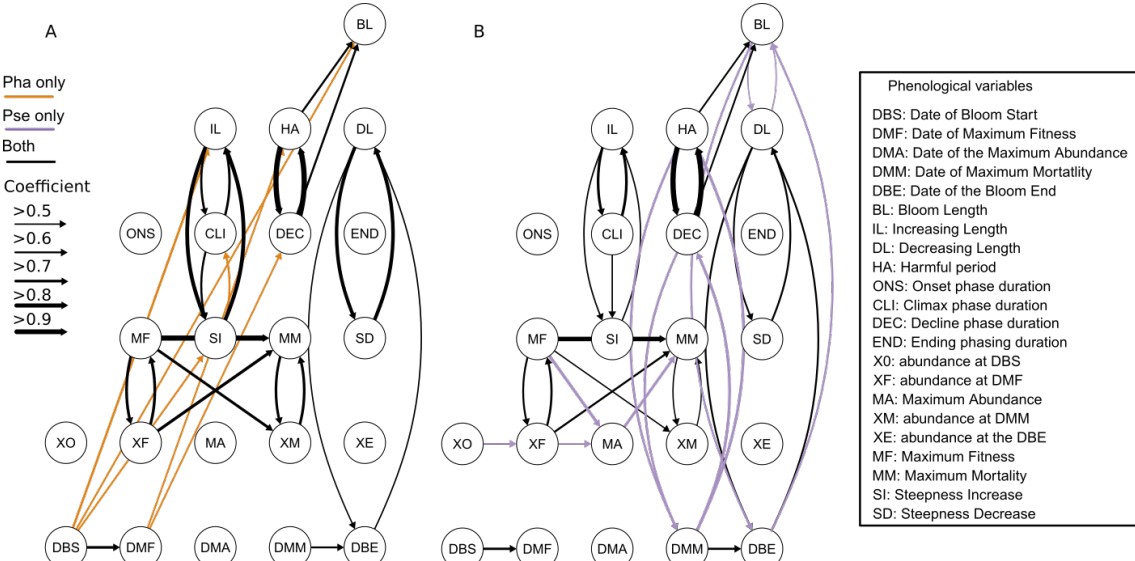

**Figure 7.** Summary of the pairwise qad relationship between the phenological variables characterizing the bloom of *Phaeocystis* sp. in (**A**), and of the complex *Pseudo-nitzchia* in (**B**). Only the significant (*p* > 0.05) and a quantification of dependence >0.5 are shown. All coefficients with their respective significance can be found in the Figure S3.

For Pha, the timing of the bloom start (DBS) affected the increasing phase via its length (IL:0.60) and slope of increase (SI:0.55) which in turn impacted on the length of the Climax phase (SI to CLI: 0.57) (Figure 7A). Consequently, the DBS of Pha has an influence on its entire bloom length (BL 0.55) (Figure 7A). In addition, both the HA and DEC bloom phases appeared affected by DMF (HA:0.55 and DEC:0.55). For Pse, the DMM was related to the HA (0.66) and DEC (0.66) (Figure 7B). Similarly, the DBE defined the end of the bloom influenced the BL (0.60) (Figure 7B). The weak interdependence between DL and BL (0.5) implied the subtle implication of the DL on the BL and vice versa (Figure 7B). Additionally, the MA also appeared to be related to the MF (0.68) and in turn impacted the MM of the bloom (0.65). Other weaker dependencies between the abundance variables were also significant (XO->XF: 0.53 and XF->MA: 0.51).

*3.5. Phenological Subsets*

The five subsets (colored contour in Figure 8) representing the environmental conditions at each phenological stage were all significant (*p* < 0.05) (Table 5). For the taxa, the highest number of samples was for the HAB subsets, which by construction was equal to the CLI and DEC subsets (Table 5). The HAB subset of Pha was, in the majority, composed of samples from the DEC (n = 718) subset rather than CLI (n = 292), whereas for the Pse it was almost equivalent (480 and 540 for CLI and DEC respectively). The subset with the lowest number of samples was ONS for Pse and CLI for Pha (Table 5). The number of samples by subsets did not reflect the size of their respective K in Figure 8. For instance, the smallest polygon for Pha and Pse were the subset ONS (the second smallest n with 317) and DEC (second highest n count with 540) respectively (Table 5). On the other hand, the largest subsets for both taxa were HAB (60.1 and 43.4 for Pha and Pse respectively), which both had the most samples (1010 and 1020 for Pha and Pse, respectively). Despite all K subsets being significant, not all mean environmental values were considered to be statistically different for the overall mean value (E) (Table 5). The non-significant values prevent the over-interpretation of the effect of the environmental variable within the subsets in which it is not significant.

**Table 5.** Environmental parameters mean value and significance by phenological subset K: onset (ONS), the climax (CLI), the harmful stage (HAB), the decline (DEC) and the end (END) for each of the harmful taxa; *Phaeocystis globosa* (Pha) and the complex *Pseudo-nitzchia* spp (Pse). The number of samples by subset K corresponds to n and the significance is of $p < 0.05$ represented by the *. A is the area of the subset K polygon in Figure 8.

| Taxa | K (n) | CHLO | SPM | $NH_4^+$ | NO | PAR | $PO_4^{3-}$ | SALI | $Si(OH)_4$ | TEMP | TURB | A |
|---|---|---|---|---|---|---|---|---|---|---|---|---|
| Pha | ONS (317) | 1.7 * | 13.3 * | 1.2 | 21.5 * | 79.2 * | 0.7 * | 33.4 * | 8.4 * | 8.1 * | 9.4 * | 26.8 |
| | CLI (292) | 6.1 * | 6.9 | 0.8 * | 13.4 * | 153.6 | 0.4 * | 33.6 | 3 * | 8.2 * | 6.2 | 30.2 |
| | HAB (1010) | 8 * | 9.7 | 0.9 * | 8.6 | 183.5 * | 0.4 * | 33.6 * | 3.2 * | 11 * | 7.8 * | 60.1 |
| | DEC (718) | 8.8 * | 10.8 | 1 * | 6.7 * | 195.7 * | 0.4 * | 33.5 * | 3.3 * | 12.2 * | 8.4 * | 58.6 |
| | END (385) | 5 | 3.9 * | 0.8 * | 2.4 * | 243.3 * | 0.3 * | 33.9 * | 1.7 * | 16.1 * | 3.2 * | 28.1 |
| Pse | ONS (283) | 2.3 * | 17.8 * | 1.2 | 20.4 * | 88.2 * | 0.7 * | 33.6 * | 8 * | 7.6 * | 11.6 * | 32.2 |
| | CLI (480) | 9.1 * | 6.9 * | 0.6 * | 7.8 * | 188.3 * | 0.3 * | 33.9 * | 2.2 * | 9.7 * | 5.9 | 41.1 |
| | HAB (1020) | 7.3 * | 5.8 * | 0.7 * | 4.8 * | 206.2 * | 0.3 * | 34 * | 2 * | 12.5 | 4.8 * | 43.4 |
| | DEC (540) | 5.7 * | 4.8 * | 0.8 * | 2.2 * | 222.2 * | 0.3 * | 34.1 * | 1.8 * | 15.1 * | 3.8 * | 29.8 |
| | END (332) | 3.7 * | 8.3 | 1.2 | 4.1 * | 188.2 * | 0.4 | 33.9 * | 4.1 | 17 * | 5.8 | 36.5 |

For Pha, 9 out of 50 mean values were not significant. The concentration of chlorophyll-*a* (CHLO) increased across the subsets from ONS (1.7 µg L$^{-1}$) to DEC (8.8 µg L$^{-1}$) (END not significant) (Table 5). Similar was observed for the mean value of PAR (79.2 to 243.3 $10^3$ W m$^{-2}$), SALI (33.4 to 33.9), and TEMP (8.1 °C to 16.1 °C) from ONS to the END. By contrast, the NO concentration decreased (21.5 to 2.4 µmol L$^{-1}$) from ONS to the END (Table 5). $PO_4^{3-}$ and $Si(OH)_4$ concentration also decreased from ONS to END but seemed to have stabilized from CLI to DEC (0.4 µmol L$^{-1}$ for $PO_4^{3-}$ and around 3.1 µmol L$^{-1}$ for $Si(OH)_4$). There was a higher SPM concentration in ONS (13.3 mg L$^{-1}$) than in END (3.9 mg L$^{-1}$) but the other mean values were not significant (Table 5). The concentration of $NH_4^+$ increased from CLI (0.8 µmol L$^{-1}$) to DEC (1 µmol L$^{-1}$) and further decreased in the END (0.8 µmol L$^{-1}$) subsets (Table 5). Finally, there was an overall decrease of TURB from ONS (9.4) to END (3.2) despite the increase between HAB (7.8) and DEC (8.4).

Now considering Pse, 8 out of 50 mean environmental values were not significant (Table 5). The CHLO concentration increased between ONS (2.3 µg L$^{-1}$) and CLI (9.1 µg L$^{-1}$) and then decreased to END (3.7 µg L$^{-1}$). A decreasing trend was observed from ONS to DEC for SPM (17.8 mg L$^{-1}$ to 4.8 mg L$^{-1}$), $PO_4^{3-}$ (0.7 µmol L$^{-1}$ to 0.3 µmol L$^{-1}$), $Si(OH)_4$ (8 µmol L$^{-1}$ to 1.8 µmol L$^{-1}$), and TURB (11.6 to 3.8). NO mostly decreased from ONS to END (20.4 to 4.1 µmol L$^{-1}$) but increased from DEC to END (2.2 to 4.1 µmol L$^{-1}$). PAR and SALI increased from ONS to DEC and finally decreased in END (Table 5). $NH_4^+$ increased (0.6 to 0.8 µmol L$^{-1}$) from CLI to DEC whereas the TEMP increased across all five subsets (from 7.6 °C to 17 °C).

The K subsets representing environmental conditions at each phenological stage were represented by the colored contour in Figure 8. The mean of the K subsets, $G_K$, were represented by the same colored triangles within the K (Figure 8). All five $G_K$ were positioned around G, which is the origin of the biplot representing the mean environmental conditions of the dataset studied (Figure 8). ONS, being the first phenological subsets (yellow triangle), was for both of them located on the left-hand side of the biplot along the x-axis (Figure 8). In contrast to ONS $G_K$ positions, the others seemed to follow the TEMP and SALI gradients. The other $G_K$ were following each other, starting with CLI higher up in the OMI plan along the y-axis and diagonally downwards to the bottom right-hand side of the OMI plan (Figure 8). The phenological mean subset conditions appeared to follow a cycle as expected with the seasonal change throughout the year.

ONS subsets were mostly defined by high nutrient concentration, turbidity (TURB) and suspended particulate matter (SPM) (Figure 8). For Pha, the subsets appeared more by restricted light (PAR) with a reduced concentration of chlorophyll (CHLO) produced by the community than for Pse (yellow contour in Figure 8). CLI subsets had shifted upwards toward conditions with higher light but lower SPM and nutrient concentration and a community producing higher chlorophyll concentration (CHLO).

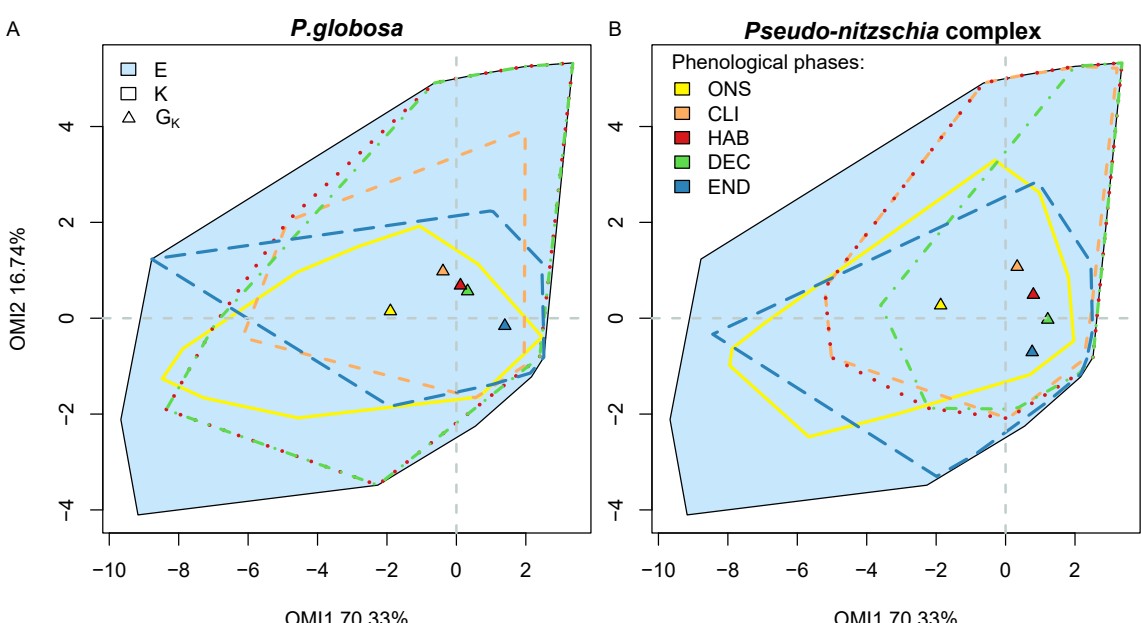

**Figure 8.** A biplot representing the distribution of the 5 environmental subsets, K, defined by the phenological stages onset (ONS: the full yellow contour), climax (CLI: dashed orange contour), harmful stage (HAB: red dotted contour), decline (DEC: green dash and dotted contour) and end (END: the blue dashed contour). Each subset has their respective mean position ($G_K$) represented by a triangle of the same color. All subsets are enclosed within the environmental space E (light blue polygon). (**A**) are the different environmental subsets for *P. globosa* and (**B**) for the complex of *Pseudo-nitzschia* spp.

Pha's subsets still appeared limited by PAR and reflected by the lower CHLO concentration produced by the community compared to Pse. HAB subsets were mostly defined by high CHLO production for both taxa. In addition for Pha, the HAB subset was also defined by high nutrient concentration, SPM, and TURB making its subsets larger than that of Pse. The CLI and HAB subset for Pse were similar in shape and size. The DEC subsets for Pha was almost the same as in HAB conditions. By contrast, the DEC subsets for Pse were smaller than in HAB conditions and were more restricted in terms of nutrient concentration, SPM and TURB. Finally, for both of them, the END subsets were mostly defined by low CHLO production by the community and high NO and TURB for Pha and high $PO_4^{3-}$, $Si(OH)_4$, $NH_4^+$, and SPM for Pse.

### 3.6. Phenological Subniche

The subniches of the two harmful algae were all significant under their respective phenological subsets ($p < 0.001$) (Table 6). For both taxa, the most marginal subniche, or the most atypical habitat used, was ONS (WitOMI$G_k$: 0.594 and 0.102 for Pha and Pse, respectively). The subniche with the lowest marginality was CLI (WitOMI$G_k$: 0.295) for Pha and HAB (WitOMI$G_k$: 0.003) for Pse (Table 6). The tolerance of Pha was greater in DEC (Tol: 2.651) compared to other subsets because the subset allowed it to be used for most of its realized niche ($N_R$) (Figure 9). The highest tolerance for Pse was the ONS subniche (Tol: 1.962) which enables it to start blooming in a wider range of environmental conditions than Pha (Table 6). The subniche with the lowest tolerance for Pha was CLI (0.48) (Table 6). The low CLI tolerance expressed its narrow range of conditions within which its bloom can actually increase. In opposition to Pse, its lower tolerance was for the DEC subniche (0.717). This result revealed that the environment restricted the bloom of Pse during its HAB phase.

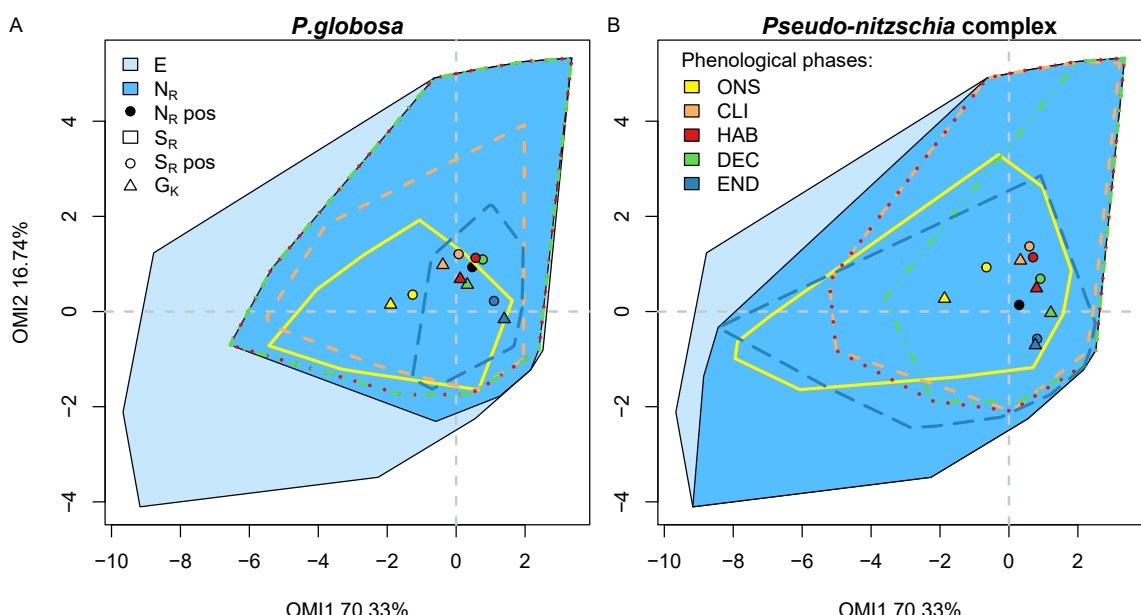

**Figure 9.** A biplot representing the distribution of the 5 phenological subniches defined by the phenological stages onset (ONS: the full yellow contour), climax (CLI: dashed orange contour), harmful stage (HAB: red dotted contour), decline (DEC: green dashed contour) and end (END: blue dashed contour). Each subniche has their mean position ($S_R$) represented by a dot of the same color. Triangles of the same color represent the mean environmental condition ($G_K$) encountered during the phenological stages. All the subniches are enclosed within the realized niche, $N_R$, position (black dot) and breadth (blue polygon) found within the environmental space E (light blue polygon) for (**A**) *Phaeocystis globosa* (Pha) and for (**B**) the complex *Pseudo-nitzchia* spp (Pse), respectively.

**Table 6.** WitOMI$G_K$ parameter significance with the $S_B$ and community variable mean values of *Phaeocystis globosa* (Pha) and the complex *Pseudo-nitzschia* spp. (Pse) in their respective phenological subsets.

| Taxa | Phase | Inertia | WitOMI$G_K$ | Tol | $S_B$ | *p* Value | H | S | J |
|------|-------|---------|-------------|-----|-------|-----------|---|---|---|
| Pha | ONS | 9.444 | 0.594 | 1.718 | 6.665 | <0.001 | 1.797 | 19 | 0.617 |
| | CLI | 5.769 | 0.295 | 1.863 | 3.438 | <0.001 | 1.253 | 20 | 0.421 |
| | HAB | 7.536 | 0.429 | 1.747 | 1.774 | <0.001 | 1.096 | 20 | 0.369 |
| | DEC | 8.035 | 0.547 | 1.677 | 1.774 | <0.001 | 1.032 | 19 | 0.348 |
| | END | 5 | 0.407 | 0.552 | 16.596 | <0.001 | 1.399 | 20 | 0.475 |
| Pse | ONS | 8.807 | 0.102 | 2.651 | 4.172 | <0.001 | 1.786 | 20 | 0.604 |
| | CLI | 7.046 | 0.005 | 1.825 | 0 | <0.001 | 0.988 | 20 | 0.329 |
| | HAB | 6.536 | 0.003 | 1.642 | 0 | <0.001 | 1.159 | 20 | 0.384 |
| | DEC | 5.246 | 0.006 | 1.174 | 0 | <0.001 | 1.311 | 21 | 0.433 |
| | END | 6.222 | 0.022 | 1.675 | 4.02 | <0.001 | 1.594 | 22 | 0.522 |

## 4. Discussion

The environmental conditions defined by the space E, within which 47 phytoplankton species niches were calculated by OMI analysis, were divided into two different environment types—estuarine and winter-like conditions, and the more open-water and summer-like conditions (Figure 4). They were suspected as the REPHY/SRN stations are mostly located in coastal waters (Figure 1), and their proximity to rivers created a salinity and turbidity gradient between the sampling, which was also found in the PERMANOVA results (Table 2). The correlation between irradiance and temperature was expected as the eastern English Channel follows the seasonal cycle which is strongly driven by these two environmental factors [76,77]. The phytoplankton community was distributed along the gradient created by nutrient concentration and light, reflecting the seasonal cycle

(Figure 4B). The strong seasonal cycle within the environmental parameters was also apparent in the results of PERMANOVA (Table 2). The segregation between the diatoms (on the left in Figure 4B) and the dinoflagellates (on the right in Figure 4B) niche positions is typical of the species seasonal succession [78]. In the EEC, the dominance shift in the succession from the diatoms to the dinoflagellates is caused by the decline of silicate concentration, which becomes a limiting factor for diatom growth. The dinoflagellates, which do not require silicate, can grow and overcome them [79,80]. The two species of interest *Phaeocystis globosa* (Pha) and the complex *Pseudo-nitzschia* (Pse) spp. preferred summer-like conditions, with Pse having a larger realized niche ($N_r$) than Pha. Pse appeared to be a much more generalist species, being more tolerant to environmental variation and found all year round (Figure 6A) compared to Pha (high tolerance and lower marginality, see Figure 5). Pha has a distinct habitat because it appeared during a time of year when there is high productivity from the community as expressed by the high concentration of chlorophyll-*a* (Figure 4B). Herein, the concentration of chlorophyll-*a* was used as a proxy for the net productivity at time *t* when the sampling occurs, as the grazing has already happened and is therefore reflected in the sampled value. To understand the relationship, the bloom of Pha needs to be put back in the ecological context. Along the French Coast of the eastern English Channel, *Phaeocystis globosa* competes with diatoms, such as *Pseudo-nitzschia* spp., for resources, especially nitrogen, phosphate and light, but only when silicate was available [35]. The "silicate-Phaeocystis hypothesis" [81,82] determined the appearance of *Phaeocystis* spp. The silicate concentration defined the duration and stability of the diatom community, which in turn determined the *Phaeocystis globosa* appearances.

One of the main parts of the study was to create an algorithm with which the blooms of the two species groups of interest, *Phaeocystis globosa* (Pha) and the complex *Pseudo-nitzschia* spp. (Pse), were detected for each time-series of each sampling station. Even with the first condition of the algorithm, which did not consider as a bloom maxima lower than 10,000 cells·$L^{-1}$, we still managed to detect a significant number of blooms for both taxa. Our expectation was to detect at least one bloom per year per taxa due to the recurrence of the HAB events in these waters [27–29,48], making an expected bloom count of 454. For a total of 363 blooms detected, 80% of our expectations were met, despite some encountered issues. For some time-series (see Figures S1 and S2 in Supplementary Materials), the estimated smooth spline at the beginning was not adequate for the detection of a bloom during the first years of the time-series. This issue was frequent and could be addressed in the future by imposing a maximum or a minimum starting point value for the estimated smooth spline. The other problem encountered was mostly related to taxonomic data. The algorithm did better at detecting the bloom of Pha, where the spline generated high peaks in a short time span, compared to Pse, which was not as clear. We think this was due to the taxa Pseu, which group several species of *Pseudo-nitzschia* spp. instead of mostly one for Pha. The *Pseudo-nitzschia* spp. complex from the REPHY/SRN dataset group, in the EEC, at least three species (*P. delicatissima*, *P. pungens*, *P. fraudulenta*) which can only be identified by closer examination of their morphology with a Scanning Electron Microscope (SEM) [31]. The diversity of unrecognized species in Pse caused an overall constant presence of the taxa and therefore generated large blooms that did not fit the 2nd and/or 3rd condition for them to be considered further. For future use of the algorithm, the taxa studied should avoid species grouping if possible and tweak the algorithm conditions (minimum abundance threshold for bloom, percentage of difference between high and low points, days of extended bloom) for each taxa instead of using the same for all of them.

From the bloom detected with the algorithm, a total of 22 phenological variables were extracted from each bloom for the two taxa giving a detailed description and quantification of the bloom characteristics. A Kruskal–Wallis Rank Sum test revealed that for most of the phenological variables, the two taxa Pha and Pse had significantly different blooms (Figure 6). These results were expected as the two taxa are from two distinct groups of phytoplankton—*Phaeocystis* is a marine haptophyte (Lancelot et al., 1994) and the complex

*Pseudo-nitzschia* is a cacillariophyte [17]—and would not have the same interaction with the environment or resource requirements. This difference in requirement can be seen in the evolution of the phenological subset K (Figure 8). At the beginning, the onset (ONS) subset was relatively similar for the two species (yellow polygon in Figure 8) but their subniche was quite different (yellow polygon in Figure 9). Pse had a greater affinity to high concentrations in nutrients, and most likely $Si(OH)_4$, than Pha and, for similar values of diversity indexes, Pha was more affected by biological interactions than Pse (Table 6). The high concentration in $Si(OH)_4$ prevented Pha from realizing its subniche properly due to competition with the diatoms for the other resources. However, Pse was more suited to compete as it was favored by this type of habitat. The high competitiveness community during the ONS phase of the bloom was also supported by high diversity indices, which indicated a relatively even community (J) with a high diversity (H) (Table 6).

Later, when the concentration of $Si(OH)_4$ reduced across the climax (CLI), harmful (HAB) and decline (DEC) subsets, this was when Pha actually managed to increase the size of its subniche as the environment changed to favor its growth despite some biological constraints (Figure 8). Similarly for Pse, this was when the taxa managed to fully occupy its potential subniche (Table 6). Despite the same number of species, the diversity indices decrease as the remaining taxa in the community were more dominant. A lower phosphate concentration allowed *Phaeocystis globosa* to out-compete the diatoms, as *Phaeocystis globosa* has the capacity to store phosphates within its colony matrix [83,84] coupled with its lower P requirements [85]. Additionally, *Phaeocystis globosa* has a strong competitive ability to obtain nitrogen [86], along with a lower concentration of silicate, and inhibited the diatom community from blooming. Since the *Pseudo-nitzschia* spp. Complex is an assemblage of species, its presence over other taxa can be for multiple reasons. In the waters of the eastern English Channel, the *P. seriata* complex is known to be abundant when low DIN and $Si(OH)_4$ concentrations result in low Si/N ratios [31]. Furthermore, limitations in silicate have been shown to favor *Pseudo-nitzschia* abundance and toxicity [87,88]. *P. multiseries* was able to out-compete other diatom species when silicate concentration was limited compared to nitrogen [89]. Therefore, in similar environmental conditions, Pha and Pse can out-compete other diatom species of the community. It was also during the CLI, HAB and DEC phases of their respective bloom that the species subniche positions were the closest to the realized niche positions, and therefore the most harmful. Finally, in the END phase, the two taxa were prone to higher biological constraints as the environment changed in favor of higher diversity (Table 6), reducing the subniche of Pha and Pse. During the END phase of the bloom, Pha and Pse were more subject to biological restrictions (Table 6). Overall, Pse appeared to be more limited by environmental conditions than restricted by biological interactions. However, Pha was always competing for resources, regardless of whether the environment was limiting or not.

The blooms of the two taxa started at approximately the Date of Bloom Start (DBS) and Date of Maximum Fitness (DMF). Pha had a short but intense burst in abundance expressed by its small Bloom Length (BL), high Steepness Increase (SI) and low Steepness Decrease (SD) with a high Maximum Abundance (MA), while Pse had a more softened (lower SI and higher SD) bloom which lasted longer (longer BL) and at lower abundance (reduced MA) (Figure 6). For a similar DBS and DMF, the two taxa appeared to have two different phenological characteristics, which had led to different phenological strategies (Figure 7). For both of them, the DBS and DMF cannot be used to determine the other key dates of the bloom, in particular the Date of the Maximum Abundance (DMA), which could be a useful bloom variable to predict for coastal resource management [13]. Furthermore, the lack of relationships between dates and abundance variables imply that the processes that are influencing the bloom timing (dates) and intensity (abundance) are different (Figure 7). Most likely, environmental conditions define the timing, and the biological interactions determine the abundance.

The taxa's bloom length (BL) appeared to be more related to the length of the HAB and DEC phase of the bloom than to the earlier phases. Moreover, the length of the HAB phase

would mostly depend on the length of the DEC than on the CLI phase (Figure 7). These links suggest that the duration of the bloom would depend on the time it takes for the corresponding taxa to decline after being established in the water column. The bloom of Pha and Pse would therefore be limited in time by external factors, either environmental [90–92] or biological [41,93,94]. It has already been reported that the spring bloom ends when the resources become limited or when the population of the grazers become sufficient to reduce the bloom size [95]. *Phaeocystis globosa* is well known to change life forms, single cells and colonies in response to grazing [96]. It responds to different chemical cues released by different predator species and is capable of switching from single cells to colonies when grazed by ciliates [97,98]. Inversely, grazing copepods can significantly decrease *Phaeocystis* spp. and its colony numbers by 60–90% [97]. However, what is unknown is the impact these limiting factors have on the duration of the entire bloom from start to end. The increasing length (IL), which consisted of the ONS and CLI duration, would have happened as long as the conditions were right for the taxa to bloom. The bloom would increase indefinitely if it were not for other environmental and/or biological factors to intervene to stop the increasing phase and start the DEC. Therefore, the efficiency of the limiting factor, in this case predation by copepods for Pha, would define the bloom duration of the taxa. The study of the environmental and/or biological factors occurring during the DEC are paramount if one wants to find a potential environmental and/or biological tools to reduce the effect of HAB.

The relationship between Maximum Fitness (MF), abundance at DMF (XF), Maximum Mortality (MM) and abundance at DMM (XM) suggested a symmetry of the blooms. In suitable environmental and biotic conditions, the abundance XF would intimately affect the MF or growth rate of the population and vice versa (Figure 7). In unsuited environmental and/or biotic conditions, the abundance XM would intimately affect the MM of the population and vice versa. The more abundant the population (XF), the higher the number of individual in the next generation, which would also reproduce and consequently increase the growth rate (MF) (Figure 7). By contrast, during the decreasing phase of the bloom, the more abundant the population XM, the higher the mortality (MM) in the population as more individuals perish due to unfavorable environments or by lethal interaction with other organisms. During the increasing phase, a positive loop exists between XF and MF whereas during the decreasing phase, a negative loop exists between MM and XM. Furthermore, the initial speed of increase (MF) and abundance (XF) has a direct impact on future mortality rate (MM) of the population (Figure 7). We explain these correlations to be similar to the process between the XM and MM but with higher temporal lag. The intensity of the positive loop during the increase would eventually affect the MM when the environmental and/or biotic conditions become less suitable to the population later in the year. The balance between the positive and the negative feedback loop during the increasing and decreasing phase of the bloom creates the known inverse U-shape characteristics of the bloom in the time-series.

Despite the common correlation between the phenological variables that the two taxa had, they also had some links that are unique to each of them. For Pha, the DBS has a direct effect on the IL, BL and SI (Figure 7A). The DBS appeared important for Pha, as not only did it mark the start of bloom, but it would also determine the duration of the bloom with its SI and therefore affecting the duration of the IL. To understand the relationship, it is necessary to put the bloom of Pha back in the ecological context. As previously mentioned, in the eastern English Channel, the appearance of *Phaeocystis* spp. was governed by the "silicate-Phaeocystis hypothesis" which competes for nutrient resources with the diatoms [81,82]. A previous study showed that the maximum in N:Si corresponded to the start of the *Phaeocystis* spp. bloom [35,99,100] giving a short window for the taxa to out-compete the diatoms. The reduced concentration of silicate represents a short environmental opportunity for *Phaeocystis* to bloom fast and out-compete the diatoms, which make the DBS a critical phenological variable. Similarly, we suspected the connection between DMF and the two bloom phases HAB and DEC was potentially

caused by mismatching (Figure 7A). As we mentioned previously, the HAB, and more specifically the DEC phase duration, would appear to depend on the efficiency of the limiting factors and/or the biological interactions to reduce the taxa population during its bloom. Therefore, the timing at which Pha reaches its highest fitness potential (DMF) is crucial for taxa life history events (such as bloom) in a seasonal cycle [101–103] as it allows it to eventually avoid competition and/or predators [43]. The timing of life history events (such as the maximum growth or MF) in a seasonal cycle can have an important fitness consequence [43], in this case the duration of the bloom.

The relationships between the phenological variables were more numerous for Pse and could potentially be used to build a predictive model. From the 22 phenological variables, only three were left unbound to another variable. Moreover, from the 32 significant links, only one was isolated from the rest, which expresses great interconnectivity between all the phenological variables. The bloom of Pse seemed to be characterized by two systems. The first system shaped the abundance of the bloom: from the start (XO) to the maximum mortality (XM) by passing through the bloom balance process (the relationship between MF, XF, MM and XM). The continuous influence of three consecutive abundance variables (XO->XF->MA) could potentially lead to helping to create a statistical tool for predicting the intensity of the bloom maximum (MA), which would be useful for coastal resource management. The second system was defined by the timing of maximum mortality and end of bloom (DMM and DBE), affecting the HAB and DEC phase as well the BL. The independence of the second system from the first suggested that Pse has no control over its decreasing phase of the bloom, which is most likely being caused by an environmental or biological limiting factor.

The relationships between the SD, DL and DBE can be explained by the way the variables are obtained (See Equation (5)). The calculation of SD depends on DL which in turns was obtained with DBE, which in turns creates the dependencies between the three variables. A similar explanation can be provided for the interdependencies between IL and SI (See Equation (4)). These explanations can only be valid because in both cases the abundance at the bloom start (XO) and at the end (XE) were constantly much lower than at the MA, increasing the dependencies between the steepness variables (SI and SD) with the tendency length (IL and DL) (Figure 6). This was verified by the stronger relationship between IL and SI for Pha than for Pse (Figure 7). It would not have been the case if XO, XE and MA had much greater variations in their respective values. By construction, the direct relationships between SI, MA, XO and IL and between SD, MA, XE and DL are suspected if XO, XE and MA were not as constant.

In perspective, this kind of study can be repeated on other harmful species found in the different regions, such as France, in *Alexandrium minutum* in the Bay of Brest [104], *Lepidodinium chlorophorum* in the Bay of the Vilaine [105] and also *Ostreopsis ovata* in the Meditterean Sea [106]. Each local area could have a statistical map representing the ecological niche of the phytoplankton community, and tracking the environment changes within the map. For each harmful species found locally, an investigation can be done to understand the impact of the seasonal change upon the phenological phases of bloom to develop biological and statistical indicators that could help coastal resource management to act upon the threat of local HAB. As the spatial scale should remain relatively small (regional size at most), the temporal resolution of the data could be increased. In our study, we used the data collected by the REPHY/SRN monitoring network [49,50] that is collected once or twice a month. Samples collected at fixed stations are often collected bi-weekly, weekly, or even daily in some monitoring programs, which exposes the detailed changes in phytoplankton variability [2]. The temporal resolution of the data is paramount for phenological analysis. Phenological studies would gain in detail with a higher frequency data, as more information would give a more precise characterization of the bloom along with the environmental variables. The phenological signal can be better predicted when observed at a fixed station. It will be necessary to identify the year-to-year variability of the source region for the population of interest [43]. Spatially explicit Lagrangian tracking

could be used to provide detailed information on the time and space scales of processes relating to the observed phenology signals. A simple passive particle backtracking shows the variability of advection pathways and source regions (e.g., [107]). To better disentangle the contribution of advection and population dynamics, backtracking for both physical and biological processes will be helpful [108,109].

## 5. Conclusions

Spatial–temporal variation performed over environmental and community data revealed a strong influence of seasonal changes, with a rather spatially homogeneous habitat across stations and from year to year. Stations differed in terms of salinity and turbidity, which was explained by the distance of sampling from the shore and from rivers. The Outlying Mean Indexes revealed that the distribution of the community ecological niche mostly depends on taxa affinity to nutrient concentration, temperature and light, which is typical of seasonal water. Diatoms were dominant in nutrient-rich waters, whereas the dinoflagellates and other family taxa were related to lower nutrients concentration and summer-like conditions. The complex *Pseudo-nitzchia* (Pse) had a large niche, whereas *Phaeocystis globosa* (Pha) had a narrower niche associated with high chlorophyll production. The bloom detection algorithm worked well—363 blooms were detected, with a higher number of blooms for Pha than for Pse. Bloom characterization by the 22 phenological variables revealed that despite having a similar bloom start, the two taxa distinguish themselves with different bloom characteristics. Pha had a short bloom length but with a steep increase and decrease and a high maximum abundance. On the other hand, the blooms of Pse were longer, less abundant and smoother. The pairwise qad relationships revealed distinct mechanisms during the blooms of the two taxa. The two taxa had in common a mechanism for bloom symmetry corresponding to the relationship between Maximum Fitness (MF), abundance at DMF (XF), Maximum Mortality (MM) and abundance at DMM (XM). Furthermore, processes causing the end of the bloom were independent of the one initiating the bloom for both. The distinction between the two blooms can be seen in their respective increasing mechanism. The blooms of Pha were more controlled by the timing with Date of Bloom Start (DBS) and Date of Maximum Fitness (DMF), whereas the increasing phase of Pse was regulated by its abundance at the different stages. The subsets of environmental conditions at the different phenological stages were similar for the two taxa, which seemed to be caused by the seasonal decrease in nutrient concentration and increasing chlorophyll production by the community and increasing temperature and light intensity. For the two taxa, their mean subniche position gravitated around the mean realized position, being the most harmful when closest to it. Furthermore, Pha appeared to be more controlled by biotic interaction (competition and/or predation), than being limited by environmental conditions. By contrast, Pse seemed to be restricted by the changing seasonal environmental conditions.

**Supplementary Materials:** The following are available online at https://www.mdpi.com/article/10.3390/jmse10020174/s1, Figure S1: Time-series of *Phaeocystis globosa* abundance for each station and with their respective detected blooms (XE = yellow, XF = turquoise, MA = red, XM = purple and XE = pink). Figure S2: Time-series of *Pseudo-nitzchia* complex abundance for each station and with their respective detected blooms (XE = yellow, XF = turquoise, MA = red, XM = purple and XE = pink). Figure S3: Hetmap representing the qad estimator between each phenological variables for the two taxa. The gray square correspond to the impossible relationship due to temporal continuum and the bold values with the $^*$ t represent the significance of the relationships. Table S1: Summary of the Kruskall-Wallis statistics of the phenological variables between the two taxa.

**Author Contributions:** Conceptualization, S.K. and A.L.; methodology, S.K.; A.L.; software, S.K.; validation, S.K. and A.L.; formal analysis, S.K.; investigation, S.K.; resources, A.L.; data curation, A.L.; writing—original draft preparation, S.K.; writing—review and editing, S.K. and A.L.; visualization, S.K.; supervision, A.L.; project administration, A.L.; funding acquisition, A.L. All authors have read and agreed to the published version of the manuscript.

**Funding:** This work has been carried out through the project S3-EUROHAB (Sentinel-3 products for detecting EUtROphicationanf Harmful Algal Bloom events) funded by the European Regional Develoment Fund through the INTERREG France-Channel-England. Futhermore, this work has also been financially supported by the European Union (ERDF),the French State, the French Region Hauts-de-France and Ifremer, in the framework of the project CPER MARCO 2015–2021.

**Institutional Review Board Statement:** Not applicable.

**Informed Consent Statement:** Not applicable.

**Data Availability Statement:** The processed data used and generated in this study are openly available in https://www.seanoe.org/data/00739/85097/ accessed on 1 January 2021 at https://doi.org/10.17882/85097.

**Acknowledgments:** The authors want also to express their gratitude to the Ifremer' LER/BL team: Devreker D., Blondel C., Duquesne V., Lebon F., Chedot B. for technical support and Ifremer' VIGIES Department for data management. We sincerely thank the Sport Nautique Valericain, Aquamarine and the Haut de France Region's crew members of vessels involved in the collection of samples at sea.

**Conflicts of Interest:** The authors declare no conflict of interest.

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
