# Peer review of "Environmental Impact on Harmful Species Pseudo-nitzschia spp. and Phaeocystis globosa Phenology and Niche"

_jmse, doi:10.3390/jmse10020174_

Round 1

Reviewer 1 Report

This is high value publication, provided by solid material and statistics. Method showed by authors can be used as a pattern for the same research. My few notes are touching publication design only.

  1. I think, that Figure A3 from the supplement should not be named as A3 but by another letter - its content is quite different from figures A1 and A2. Also it would be better to change square colors as part of the numbers and especially asterisks are almost not visible.
  2. It's better not to put several big figures and tables one after another inside the text - it becomes difficult to follow text after such a large gap (Figures 4-5 and Table 3).
  3. Small spell check is needed (see attachment) 

Author Response

Reviewer 1:

This is high value publication, provided by solid material and statistics. Method showed by authors can be used as a pattern for the same research. My few notes are touching publication design only.

  1. I think, that Figure A3 from the supplement should not be named as A3 but by another letter - its content is quite different from figures A1 and A2. Also it would be better to change square colors as part of the numbers and especially asterisks are almost not visible.

The name of the figure A3 were modify into figure B1, the color were modified so to provide more contrast between values. As we couldn’t  modify the size of asterisks without affecting the entire graph, the font of the significant values were change to bold for the sake of clarity. Also as the figure is provided in PDF, it is easy to zoom into the figure to increase its clarity.

  1. It's better not to put several big figures and tables one after another inside the text - it becomes difficult to follow text after such a large gap (Figures 4-5 and Table 3)

The figure 4 has been moved upward, but the Latex software somehow does not allow me to move figure 5 and the table 3 without affecting the rest of the text. 

  1. Small spell check is needed (see attachment)

The small spell check were done. "taxa" was not replaced by "species" as we want to keep homogeneity throughout the paper. 

Reviewer 2 Report

I enjoyed reading this manuscript the entire time. This study covers a long-time question that has been raised since Hutchins’s ‘the Paradox of the plankton’ was published. It is an ecological niche among co-existing species. Although the current study investigated the phenological characteristics of two successive bloom species, Pseudo-nitzschia spp. and Phaeocystis globosa, this work would excellently explain the ecological niche of phytoplankton. Despite the intensive data analysis and sample collection, I recommend a few more things that the authors would want to revisit.

Major revisions

  1. I understand the authors focused on ‘bloom phenology’ but why don’t the author consider the shift in bloom phenology during the sampling periods? Given that the long-term analysis, there could be a variation in phenology over time. Year-to-year variation or decadal variation would be interesting.

  1. Line 666-670 Productivity and chl a have a different story because productivity indicates the ability of phytoplankton to grow and chl a potentially indicates phytoplankton biomass. Without comparison in primary productivity between two species, it is difficult to define it. The silicate-Phaeocystic hypothesis sounds interesting but I wonder if zooplankton grazing contributed to determining the species niche, meaning that there might be less grazing on Pha than that on Pse, thereby leading to a significantly positive correlation between chl a and Pha. Secondly, Pha might physiologically contain more chl a per cell than Pse?

  1. Line 735-739 This contradicts the silicate-Phaeocystis theory.

  1. With the current phenology and niche study, where do the authors want to go? In other words, are the authors wanting to assemble a model to predict the future HABs? Where the results can be utilized/applied?

Minor revisions

  1. Materials and Methods describe that the sampling was performed since 1980s and 1990s for environmental and biological variables, respectively. However, this section does not explain when it ended.

  1. Combine section 2.2 with the previous section. Too short to be alone.

  1. Please make a table explaining three niche parameters and 22 phenological variables instead of describing all in Materials and Methods. That would be easier for readers to follow.

Author Response

Reviewer 2:

Major revisions

  1. I understand the authors focused on ‘bloom phenology’ but why don’t the author consider the shift in bloom phenology during the sampling periods? Given that the long-term analysis, there could be a variation in phenology over time. Year-to-year variation or decadal variation would be interesting.

We did consider the annual shift in bloom phenology during the sampling periods for the two species by testing the phenological variables and stations separately and combined, but it was never significant. Which is why we ended it up doing the PERMANOVA on the environmental and community variables directly to test how the monthly and yearly variation affected the data. And we notice that the annual change did not impact the environmental data, and we suspect that the annual variation was not sufficient for it to cause a significant phenological shift from year to year in the Eastern English Channel, during the time period studied. Another explanation could be that the time-period was not long enough for a shift in the environment and therefore in the bloom phenology to be observed by our analysis.  In order to satisfy futur readers with a similar question the following sentence was added in the Phenological analyses subsection of the Material & Methods explaining our first attention: The spatial-temporal variation in the bloom phenology of the two species were considered in this section in order to study potential shift in year-to-year phenology and/or spatial differentiation, but our exploratory analyses were inconclusive and therefore discarded.

  1. Line 666-670 Productivity and chl a have a different story because productivity indicates the ability of phytoplankton to grow and chl a potentially indicates phytoplankton biomass. Without comparison in primary productivity between two species, it is difficult to define it. The silicate-Phaeocystic hypothesis sounds interesting but I wonder if zooplankton grazing contributed to determining the species niche, meaning that there might be less grazing on Pha than that on Pse, thereby leading to a significantly positive correlation between chl a and Pha. Secondly, Pha might physiologically contain more chl a per cell than Pse?

We do agree with your comment, and it would be particularly difficult to measured it in situ for two species only. Productivity is not currently measured in the monitoring program yet, we only have Chla as a proxy which is strongly linked to the European directive and regional sea conventions whom assessments are based phytoplankton biomass only. They assumed that the sampled biomass represent the net productivity at the time t the sampling occurs, as the grazing has already happened and is therefore reflected in the sampled value. And to avoid any further misinterpretation of our meaning, this notion of in situ  productivity at time t was added in this part of the discussion.

To answer the second part of your comment, yes the grazing has definitely contributed to the species niche, but it is not possible to know how much the grazing pressure Pha and Pse were under. The abundance value at time t of the two species already reflect the species’ productivity and mortality. Pha could have been high grazing pressure but the sampled abundance, even if it is very high in abundance, only reflect a fraction of its original population size at time t. The main idea behind our sentence is to emphasize the fact the community productivity, at the time it was sampled, was when the abundance of Pha was high. Pse has  more Chl a per cell than Pha but herein the correlation Chla concentration is mostly due to the higher abundance of Pha than Pse as Pha can be much more abundant (see figure 6).

  1. Line 735-739 This contradicts the silicate-Phaeocystis theory.

The discussion were corrected to fit the silicate-Phaeocystis theory removing the " and therefore Pseudo-nitzschia spp." in line 676 so does not impair with the concern section. But we think that the limited concentration of silicate also potentially favored the appearance of Pse.

  1. With the current phenology and niche study, where do the authors want to go? In other words, are the authors wanting to assemble a model to predict the future HABs? Where the results can be utilized/applied?

In the last paragraph of the Discussion we mentioned a few ideas (satellite HAB tracking device, regional bloom scenarios prediction by machine learning) on how the results and the methods of our investigation can be used  in futur studies wishing to aim in predicting the occurrence of HAB in theses Eastern English Channel. As the REPHY/SRN monitoring program have for ambition to create innovative tools (models or statistical indicators) that could help in a better and more sustainable management of the coastal and marine resources, we hoped that we could provide a few leads for future investigations.

Minor revisions

  1. Materials and Methods describe that the sampling was performed since 1980s and 1990s for environmental and biological variables, respectively. However, this section does not explain when it ended.

The REPHY-SRN network is still ongoing to this date, which is why there is no end date. The sentence in line 348  was modified to clearly state the time periods of the data and well as the number of  stations. 

  1. Combine section 2.2 with the previous section. Too short to be alone.

The section was merged with the previous one. 

  1. Please make a table explaining three niche parameters and 22 phenological variables instead of describing all in Materials and Methods. That would be easier for readers to follow.

We apologize, but we don’t think that making a table would make prevent from explaining the 22 phenologicals variables, 3 niche parameters as well as the WithOMIGk parameters. The figure 2 already summarized the 22 phenological variables.  The suggested table would only repeat the text if we define each terms and would only add another large to the already extensive paper.